# Silica-copper catalyst interfaces enable carbon-carbon coupling towards ethylene electrosynthesis

Jun Li [1,2,7], Adnan Ozden[1,7], Mingyu Wan[3,7], Yongfeng Hu[4], Fengwang Li [2], Yuhang Wang [2], Reza R. Zamani [5], Dan Ren [6], Ziyun Wang [2], Yi Xu [1], Dae-Hyun Nam[2], Joshua Wicks [2], Bin Chen[2], Xue Wang [2], Mingchuan Luo [2], Michael Graetzel [6], Fanglin Che[3✉], Edward H. Sargent [2✉] & David Sinton [1✉]

Membrane electrode assembly (MEA) electrolyzers offer a means to scale up $CO_2$-to-ethylene electroconversion using renewable electricity and close the anthropogenic carbon cycle. To date, excessive $CO_2$ coverage at the catalyst surface with limited active sites in MEA systems interferes with the carbon-carbon coupling reaction, diminishing ethylene production. With the aid of density functional theory calculations and spectroscopic analysis, here we report an oxide modulation strategy in which we introduce silica on Cu to create active $Cu-SiO_x$ interface sites, decreasing the formation energies of OCOH* and OCCOH*—key intermediates along the pathway to ethylene formation. We then synthesize the $Cu-SiO_x$ catalysts using one-pot coprecipitation and integrate the catalyst in a MEA electrolyzer. By tuning the $CO_2$ concentration, the $Cu-SiO_x$ catalyst based MEA electrolyzer shows high ethylene Faradaic efficiencies of up to 65% at high ethylene current densities of up to 215 mA cm$^{-2}$; and features sustained operation over 50 h.

[1] Department of Mechanical and Industrial Engineering, University of Toronto, Toronto, ON, Canada. [2] Department of Electrical and Computer Engineering, University of Toronto, Toronto, ON, Canada. [3] Chemical Engineering, University of Massachusetts Lowell, Lowell, MA, USA. [4] Canadian Light Source Inc., University of Saskatchewan, Saskatoon, SK, Canada. [5] Interdisciplinary Center for Electron Microscopy, École Polytechnique Fédérale de Lausanne, Lausanne, Switzerland. [6] Laboratory of Photonics and Interfaces, Institute of Chemical Sciences and Engineering, École Polytechnique Fédérale de Lausanne, Lausanne, Switzerland. [7] These authors contributed equally: Jun Li, Adnan Ozden, Mingyu Wan. ✉email: fanglin_che@uml.edu; ted.sargent@utoronto.ca; sinton@mie.utoronto.ca

CO$_2$ electroreduction (CO$_2$RR), powered by renewable electricity, represents a carbon-neutral pathway for the production of value-added hydrocarbons and alcohols[1–3]. Ethylene, a precursor in the chemical industry, has been prioritized as a target in CO$_2$RR owing to its annual global production of 140 million metric tons and a market value of 182 billion USD[4]. However, the present-day performance of CO$_2$-to-ethylene electrocatalysis—in terms of efficiency, production rate, and stability—is insufficient to compete with ethylene production from fossil sources[4,5].

Gas diffusion electrodes (GDEs) embedded in alkaline flow cell electrolyzers have enabled selective CO$_2$-to-ethylene conversion at industrial-level production rates[6,7]. However, CO$_2$ utilization remains low due to the rapid reaction of CO$_2$ molecules and hydroxide ions in these systems[2]. In addition, direct contact between GDE and aqueous electrolyte leads to electrode flooding and catalyst deactivation[8,9].

The membrane electrode assembly (MEA) electrolyzer, with a direct cathode:membrane:anode contact, offers a platform that is more stable than alkaline flow cell electrolyzers[10–12]. A low cell resistance of zero-gap MEA electrolyzers also enables the use of bicarbonate or carbonate electrolyte without sacrificing the CO$_2$RR efficiency[13,14]. Copper (Cu) has an adequate activation energy of CO$_2$ and shows near-optimal CO binding, a key to producing ethylene[15,16]. However, the Faradaic efficiency (FE) toward ethylene reported on bare Cu catalysts in MEA systems is limited to <50%[10].

Excessive local CO$_2$ at the Cu catalysts surface, a result of fast mass transport of CO$_2$ in flowing gas systems, and a lack of active sites diminishes ethylene formation due to the strong lateral interaction between adsorbed CO$_2$ and CO$_2$RR intermediates[17]. Introducing an element through doping[18,19], alloying[20,21], or molecule-modification[22,23], offers a means to modify the structure of metallic Cu catalysts and has improved the FE$_{ethylene}$ to an impressive 64% in an MEA electrolyzer[22], but with ethylene productivity <100 mA cm$^{-2}$.

A recent techno-economic analysis indicates that practical and economic ethylene production requires productivities greater than 200 mA cm$^{-2}$ (ref. [2]). This indicates the need for catalysts that achieve both high selectivity (>60%) and high activity (>300 mA cm$^{-2}$), likely through designing a catalyst capable of accelerating CO$_2$ activation and enabling efficient and continuous electron/proton transfer steps with strict control over the key reaction step—the carbon–carbon (C–C) coupling[3,24].

The formation of OCCOH* intermediate from C–C coupling has been identified as a key step toward ethylene production[25]. Low-index and stepped Cu facets reduce the formation energy of the OCCOH* intermediate, due to efficient charge transfer between surface step sites and CO$_2$RR intermediates[15]. However, under-coordinated step sites are rare in bulk Cu and less stable compared to Cu(111).

Herein, we report an oxide modulation strategy based on silica—a modifier oxide that has strong affinities to Cu and O—to create abundant Cu-SiO$_x$ interface stepped sites, assisting CO$_2$ activation and the formation of the OCCOH* intermediate. We first assess the role of silica in affecting CO$_2$RR on Cu using density functional theory (DFT) calculations. We find that a silica addition on a Cu surface significantly lowers the formation energies of the key intermediates OCOH* and OCCOH* at the Cu-SiO$_x$ interface sites via forming strong Si–O or Si–C bonds, and leads to a cascade of coupled electron/proton transfer reactions that enhance CO$_2$-to-ethylene conversion. We then synthesize a Cu-SiO$_x$ catalyst using one-pot coprecipitation. Through microscopy and spectroscopy, we find that Si species uniformly distribute at the Cu surface, forming abundant Cu-SiO$_x$ interfaces to catalyze CO$_2$. We integrate the Cu-SiO$_x$ catalyst in a MEA electrolyzer and achieve a FE of 65% at a current density of 215 mA cm$^{-2}$ for CO$_2$-to-ethylene conversion, two-fold greater ethylene productivity in MEA compared to literature benchmarks. We further assess the performance of the Cu-SiO$_x$ catalyst in dilute CO$_2$ streams, with the aim of directly converting flue gases. We demonstrate stable CO$_2$-to-ethylene conversion with FEs of >60% at current densities in the range of 120–300 mA cm$^{-2}$ across a wide window of CO$_2$ concentrations (10–100%) and for a run time over 50 h.

## Results

**Density functional theory calculations.** Excessive local CO$_2$ at the Cu catalyst layer interferes with the adsorption of CO$_2$RR intermediates, diminishing the C–C coupling at the Cu surface with limited active sites for ethylene production[17]. In light of this finding, we sought to identify, with the aid of DFT calculations, a catalyst candidate having abundant active sites to enable CO$_2$ activation in MEA systems and facilitate the C–C coupling toward ethylene formation.

Si-based materials have been applied to facilitate CO$_2$ hydrogenation; however, Si itself has been found to react irreversibly with CO$_2$ by forming surface silanols (Si–OH) and siloxane (Si–O–Si) groups[26,27]. We instead considered silica, because it is more stable and is an efficient adsorbent material to capture CO$_2$[28], besides it shows strong affinities to Cu[29]. We thus hypothesized that a Cu surface modified with silica could offer a new approach to CO$_2$RR. Using DFT calculations we first introduced silica clusters at the Cu surface to establish a Cu-SiO$_x$ step surface configuration (Supplementary Fig. 1) and calculated the energetics of forming possible Cu-SiO$_x$ catalysts with various silica loadings (Supplementary Figs. 2–4 and Supplementary Table 1). We found a slight increase of the Si oxidation state from +1.9 to +2.4, as the silica concentration increases from 1/16 to 3/16 monolayer (ML) (i.e., 1.6–4.7%) (Fig. 1a).

We then investigated the first proton/electron transfer to CO$_2$ (CO$_{2(gas)}$ + H$^+$ + e$^-$ ↔ OCOH*) with and without silica, a key step for CO$_2$ activation (Supplementary Figs. 5–7). DFT results show that the Cu-SiO$_x$ catalyst significantly decreases the formation energy of OCOH* up to ~0.7 eV compared to the bare Cu catalyst, due to the strong electronic interaction between Si and O/C in the OCOH* intermediate (vide infra), indicating that a silica addition promises efficient CO$_2$ activation.

We further calculated the formation energy of OCCOH* through the coupling of CO* and COH* (Fig. 1b), a key descriptor to enhance C$_2$ production from CO$_2$RR[15,25]. We examined the co-adsorption of CO* and COH* over the energetically favorable and metastable Cu-SiO$_x$ models (Supplementary Figs. 2–4) and found that all Cu catalysts, with and without silica, exhibit similar values (Supplementary Figs. 8–12), suggesting that the silica addition does not change the adsorption properties of CO* or COH*. However, various Cu-SiO$_x$ catalysts show a much higher adsorption energy of up to ~1.2 eV for the as-formed OCCOH* intermediate than the bare Cu catalyst (Supplementary Figs. 13–15). As a result, for the Cu-SiO$_x$ catalysts with equilibrium geometries, we found that increasing the silica loading up to 3.1% decreases the formation energy of OCCOH* by ~1.1 eV (Fig. 1b). A further increase of the silica concentration to 4.7% disfavors OCCOH* formation, showing reaction energy similar to that of bare Cu. This volcano-shaped dependence of the OCCOH* formation energy based on the silica loading suggests that a proper loading of silica on Cu could provide a means to construct active interface sites for improving C$_2$.

To determine the role of silica in enhancing the OCCOH* adsorption at the Cu-SiO$_x$ step surface, we generated a differential charge density between OCCOH* and Cu-SiO$_x$ (at a silica

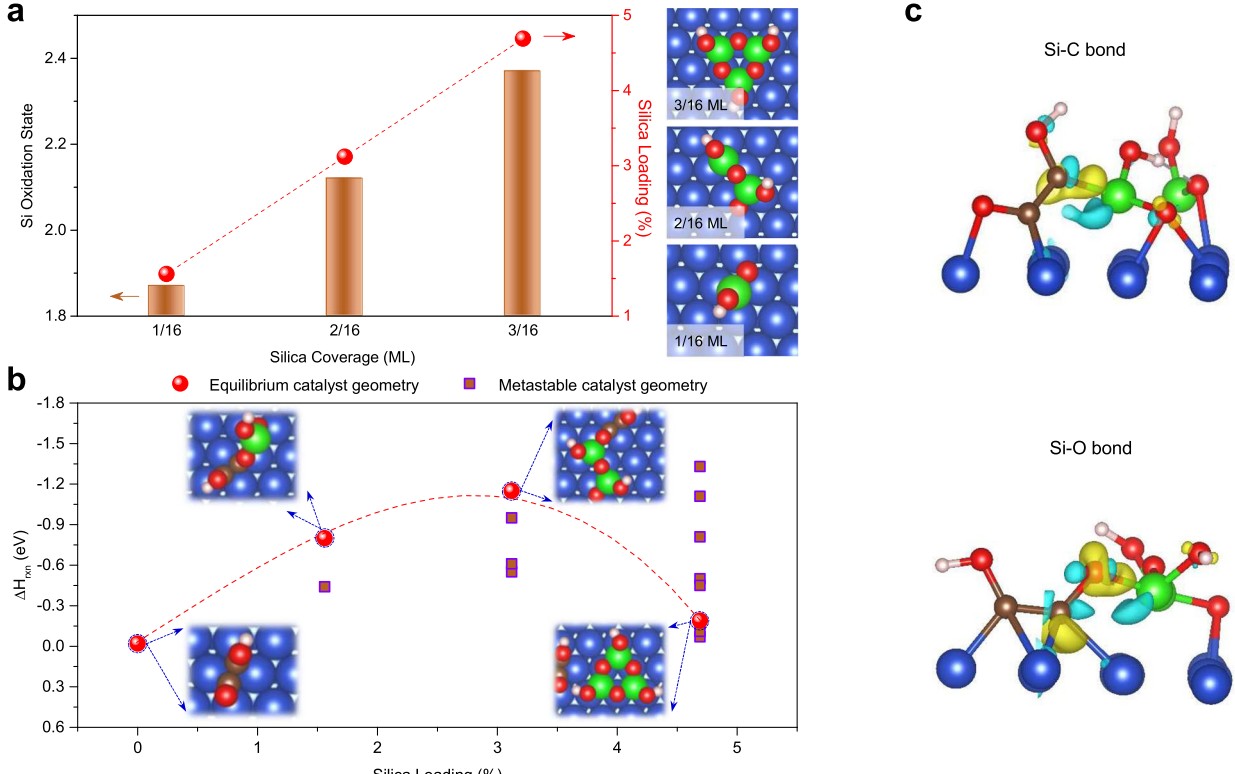

**Fig. 1 DFT calculations. a** The oxidation state of Si and silica loading as a function of silica coverage over Cu(111) at the energetically favorable catalyst geometries. **b** The formation energy ($\triangle H_{rxn}$) of OCCOH* (CO* + COH* → OCCOH*) over the pure Cu(111) and possible Cu-SiO$_x$ catalysts, including equilibrium (sphere) and metastable (square) catalyst geometries; A volcano-like plot is achieved for the formation energy of OCCOH* over the equilibrium Cu-SiO$_x$ catalyst geometries. **c** Differential charge density plots show the 3.1% silica-loaded Cu-SiO$_x$ step surface can stabilize OCCOH* adsorption by creating either Si–C or Si–O bonds and lower the formation energy of the OCCOH* intermediate. The iso-surface level of the differential charge densities is 0.012 e/bohr[3]. The yellow or blue areas represent a gain or loss of electrons. Color-coded atoms represent Cu (blue), Si (green), O (red), C (brown), and H (pink).

loading of 3.1%, Fig. 1c) in which silica loses electrons (blue) and the adsorbed OCCOH* gains electrons (yellow). For Si–C bond formation, the C$^1$ atom of the O=C$^1$–C$^2$–OH* intermediate has one electron in a $sp^2$ orbital that bonds to Cu, while the C$^2$ atom makes more of a carbene and tends to bond more strongly to Si than Cu, increasing the adsorption energy of OCCOH* and thus decreasing the formation energy of OCCOH* up to 0.6 eV (Supplementary Figs. 16 and 17). A much stronger adsorption of OCCOH* over the Cu-SiO$_x$ catalyst is due to the formation of the Si–O bond between O$^{\delta-}$ of the OCCOH* intermediate and silica at the Cu-SiO$_x$ step site, leading to an enhancement in the adsorption energy of OCCOH* and thus lowering the formation energy of OCCOH* up to ~1.2 eV (Supplementary Figs. 18 and 19).

When the silica loading is increased to 4.7%, the silica component energetically favors the formation a Si–O–Si layer, which fails to form Si–C or Si–O bond with the OCOH* and OCCOH* intermediates, due to the lack of unpaired electrons in Si. As a result, the Cu-SiO$_x$ catalyst shares similar adsorption energies of both the OCOH* and OCCOH* intermediates to the bare Cu catalyst (Supplementary Figs. 7 and 15). Collectively, the DFT calculations predict that introducing a silica addition at a loading of 3% could establish a stable Cu-SiO$_x$ interface that lowers the formation energies of OCOH* and OCCOH*, and increases the efficiency of CO$_2$-to-ethylene conversion.

**Materials synthesis and characterization**. We pursued a one-pot coprecipitation method to prepare the silica-modified Cu catalysts,

dissolving silicon tetrachloride and copper chloride in ethanol. A Cu-SiO$_x$ catalyst was then coprecipitated by adding NaBH$_4$ directly into the precursor solution. We tuned the silica loading in the Cu structure from 0 to 5% (denoted Cu-SiO$_x$-M, where M represents the silica loading percentage) by varying the Si:Cu atomic ratio in the precursor solution. The silica loading was quantified by inductively coupled plasma optical emission spectrometry (ICP-OES, Supplementary Table 2).

Transmission electron microscopy and scanning electron microscopy images show nanoparticles in an aggregated form and a mean size of ~50 nm. The Cu morphology is unchanged with silica loading (Fig. 2a and Supplementary Fig. 20). The catalysts with/without silica loading lead to similar Cu patterns, seen in X-ray powder diffraction (XRD) measurements (Supplementary Fig. 21). We examined and compared the structure of the Cu-SiO$_x$ catalyst before and after CO$_2$RR using high-angle annular dark-field scanning transmission electron microscopy (HAADF-STEM) and energy-dispersive X-ray (EDX) mapping. We found that the Si species are evenly distributed on the Cu catalyst surface, and that the structure of Si species remains intact before and after the reaction, showing an oxidation state of +2 as revealed by X-ray photoelectron spectroscopy (XPS, Fig. 2b–f and Supplementary Figs. 22–27)[30]. Using HAADF-STEM and XPS, we also characterized the Cu-SiO$_x$ catalysts with silica loadings of 2.5% (Figs. 2) and 5% (Supplementary Figs. 28–31), and found similar silica structures in both samples and detected no aggregation of silica, suggesting that silica clusters are uniformly distributed on the Cu surface. We note that characterizations of catalyst morphology were performed under ex-situ conditions.

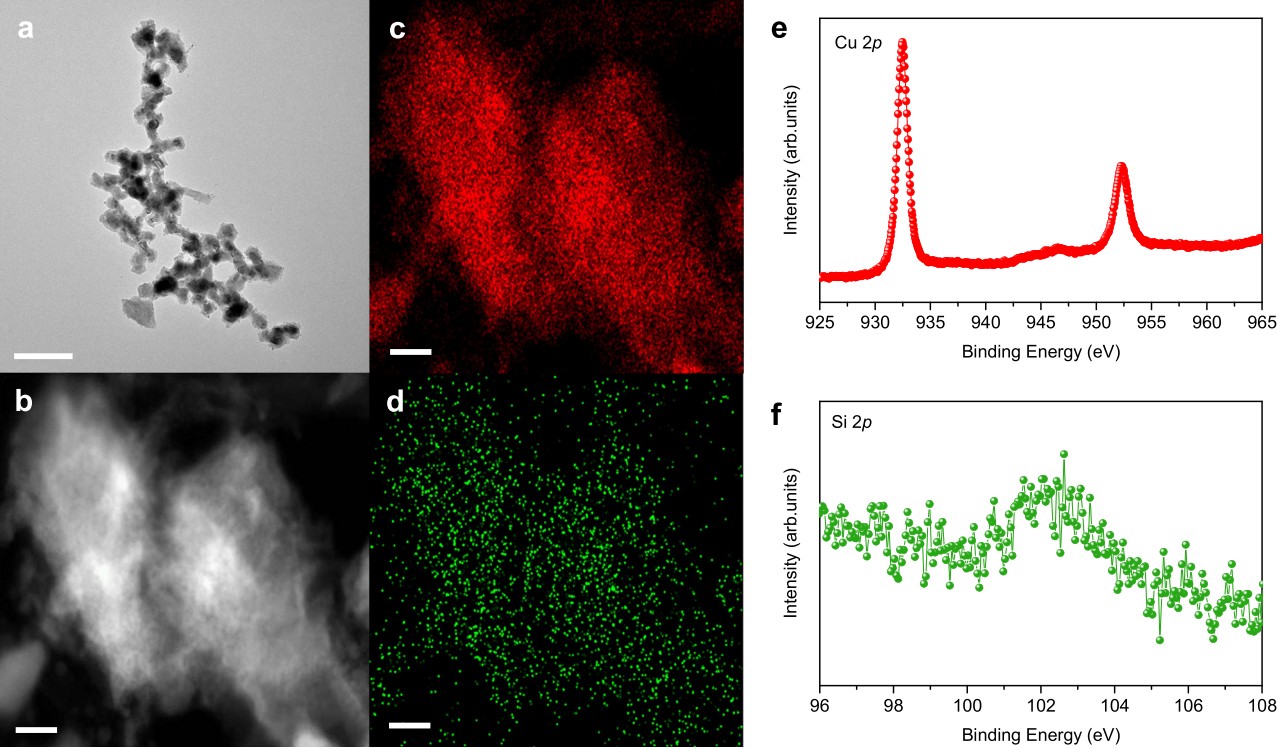

**Fig. 2 Characterization of 2.5% silica-loaded Cu catalyst. a–d** TEM image (**a**), HAADF-STEM image (**b**), and EDX elemental mapping of Cu (**c**) and Si (**d**). The scale bars are 200 nm in **a** and 20 nm in **b–d**. **e**, **f** XPS spectra of Cu $2p$ (**e**) and Si $2p$ (**f**). All measurements were performed upon completion of $CO_2RR$ at a full-cell potential of −4.1 V in 0.1 M $KHCO_3$.

Although the extent of any catalyst reconstruction due to the potential removal after $CO_2RR$ is not expected to be major, the development of in-situ TEM capabilities will be useful to assess the impact of potential removal on catalyst reconstruction, if any.

**Electrochemical $CO_2$ conversion.** We evaluated the $CO_2RR$ performance of the $Cu-SiO_x$ catalysts in a 5 $cm^2$ MEA electrolyzer (Fig. 3a), in which ethylene was produced according to the overall reaction of $2CO_2 + 2H_2O \rightarrow C_2H_4 + 3O_2$ (E° = 1.15 V). In this system, a humidified $CO_2$ feedstock was fed through the cathode without the presence of catholyte, and the $Cu-SiO_x$ catalyst of interest was sandwiched between a PTFE gas diffusion layer and an anion exchange membrane (AEM) to enable sufficient $CO_2$ reactants and abundant reaction interfaces, whereas 0.1 M $KHCO_3$ aqueous solution was interfaced with anode catalysts ($IrO_2$ coated onto a Ti mesh) to initiate the oxygen evolution reaction (OER). The ohmic loss was minimized, due to the absence of catholyte and direct contact between the catalyst and the AEM.

In a silica loading range of 0–5% on Cu, we found a strong correlation between the silica loading and ethylene formation (Fig. 3b and Supplementary Fig. 32). We observed a gradual increase in $FE_{ethylene}$ and a decrease in $FE_{CO}$ with increasing silica loading up to 2.5%, whereas we noted a decrease in $FE_{ethylene}$ with a further increase in the silica loading from 2.5 to 5%, accompanying an enhancement in $FE_{CO}$. The volcano relation between $FE_{ethylene}$ and the silica loading agrees well with the volcano correlation between the formation energy of OCCOH* and the silica loading ranging from 0 to 5% (equivalent to a silica coverage of 0–3/16 ML) predicted by DFT calculations (Fig. 1b). We observed that the total and hydrogen current densities do not vary significantly in the range of silica loadings tested (Supplementary Fig. 33), implying similar physical availability (i.e., mass transport) of $CO_2$.

We then focused on the 2.5% silica-loaded Cu catalyst ($Cu-SiO_x$-2.5) and observed that $FE_{ethylene}$ of the $Cu-SiO_x$-2.5 catalyst outperforms that of the bare Cu catalyst across the full-cell potential range of −3.0 V ~ −4.25 V (Fig. 3c). The $Cu-SiO_x$-2.5 catalyst reaches a peak $FE_{ethylene}$ of 65% at a cell potential of −4.1 V, whereas a peak $FE_{ethylene}$ of 50% of the bare Cu catalyst is achieved at −4.2 V. The ethylene current densities of the $Cu-SiO_x$-2.5 and bare Cu catalysts increase with increasing cell potential in the negative direction and peak at 215 and 160 mA $cm^{-2}$ at cell potentials of −4.2 and −4.25 V, respectively (Fig. 3d). This performance exceeds prior MEA reports with a two-fold enhancement in ethylene current density (Supplementary Table 3). The $Cu-SiO_x$-2.5 catalyst shows considerably lower $FE_{CO}$ and CO current density than the bare Cu catalyst (Fig. 3e and Supplementary Fig. 34), in agreement with our DFT results that the silica addition facilitates the C–C coupling reaction (Fig. 1). A further analysis of liquid products (Fig. 3f), using [1]H nuclear magnetic resonance ([1]H-NMR) spectroscopy, reveals that the $Cu-SiO_x$-2.5 catalyst also yields lower FEs toward multi-carbon oxygenate products (i.e., acetate, ethanol, and 1-propanol) than the bare Cu catalyst.

**Investigating the $Cu-SiO_x$ interaction.** To understand the role of silica in promoting ethylene production at the $Cu-SiO_x$ catalyst, we performed in-situ Raman spectroscopy (Supplementary Table 4). At a range of applied cell potentials from −1.5 to −2.0 V (Fig. 4a, b), we observed one band at 280 $cm^{-1}$ on the bare Cu electrode starting from −1.8 V, attributable to the Cu-CO frustrated rotation at metallic Cu surface[31,32]. On the $Cu-SiO_x$-2.5 electrode, the same band emerged at the cell potential of −1.8 V, together with the emergence of a new band at 365 $cm^{-1}$. This additional band is due to the Cu–O stretch at the metallic Cu surface, indicating that a silica addition increases CO coverage at

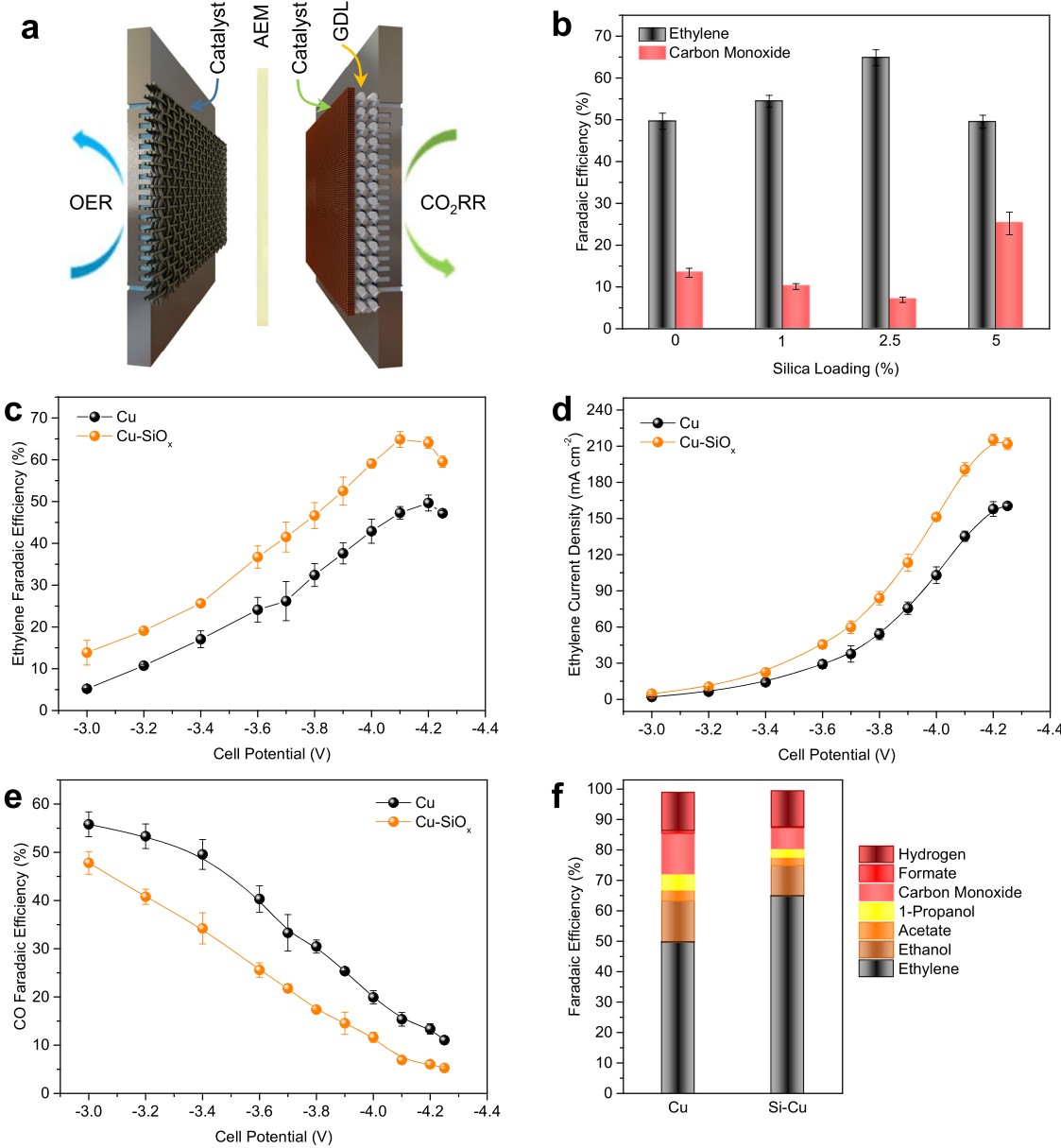

**Fig. 3 CO₂RR performance of different Cu-SiO_x catalysts in a MEA electrolyzer with 0.1 M KHCO₃ anolyte. a** Schematic illustration of the MEA cell (OER: oxygen evolution reaction, AEM: anion exchange membrane, GDL: gas diffusion layer). **b** The peak FE_ethylene and corresponding FE_CO of the bare Cu and various Cu-SiO_x catalysts. **c–e** Comparison of FE_ethylene (**c**), ethylene current density (**d**), and FE_CO (**e**), respectively, between the 2.5% silica-loaded Cu and bare Cu catalysts. **f** The peak FE_ethylene and related FEs of the other products (i.e., ethanol, acetate, 1-propanol, CO, formate, and hydrogen) of the 2.5% silica-loaded Cu and bare Cu catalysts. Error bars are means ± SD (n = 3 replicates).

the Cu surface[32]. In addition, we also observed a strong band at 530 cm⁻¹ on the Cu-SiO_x-2.5 electrode throughout the cell potentials applied, which we attribute to the presence of Cu⁺ promoted by the Cu–O–Si interaction[31].

To further assess the role of silica, we carried out in-situ Raman measurements on the Cu-SiO_x-1 and Cu-SiO_x-5 electrodes (Supplementary Fig. 35). While we observed the stable Cu–O–Si i band at 530 cm⁻¹ in both cases throughout the cell potentials applied, the Cu–O stretch band at 365 cm⁻¹ emerged at a cell potential of −2.0 V on Cu-SiO_x-1 and −2.2 V on Cu-SiO_x-5. This shift in onset potential of the 365 cm⁻¹ band indicates the tuning of CO content on Cu in response to silica loading. The Cu-SiO_x-2.5 catalyst shows the lowest onset potential of −1.8 V, suggesting the highest CO concentration at the Cu-SiO_x-2.5 catalyst surface during CO₂RR. This finding is consistent with our DFT

predictions of the formation energy of OCOH*—an intermediate in CO₂-to-CO conversion—at the Cu-SiO_x interface (Supplementary Figs. 5–7). The Cu-SiO_x catalyst with optimal CO coverage at a silica loading of 2.5% accelerates the C–C coupling by lowering the formation energy of OCCOH* at the Cu-SiO_x interface, facilitating the formation of ethylene (Fig. 1b)[33].

To reveal the electronic interaction between Cu and silica, we conducted X-ray absorption spectroscopy (XAS) measurements at the Cu K- and Si K-edges of the Cu-SiO_x catalysts. In-situ X-ray absorption near-edge structure (XANES) at the Cu K-edge shows the metallic nature of Cu active sites, corroborated by the extended X-ray absorption fine structure (EXAFS, Supplementary Fig. 36a–c). Although the presence of Cu⁺ from the Cu–O–Si interaction is evidenced by in-situ Raman analysis, the Cu K-edge XAS recorded in fluorescence yield probes further into the bulk

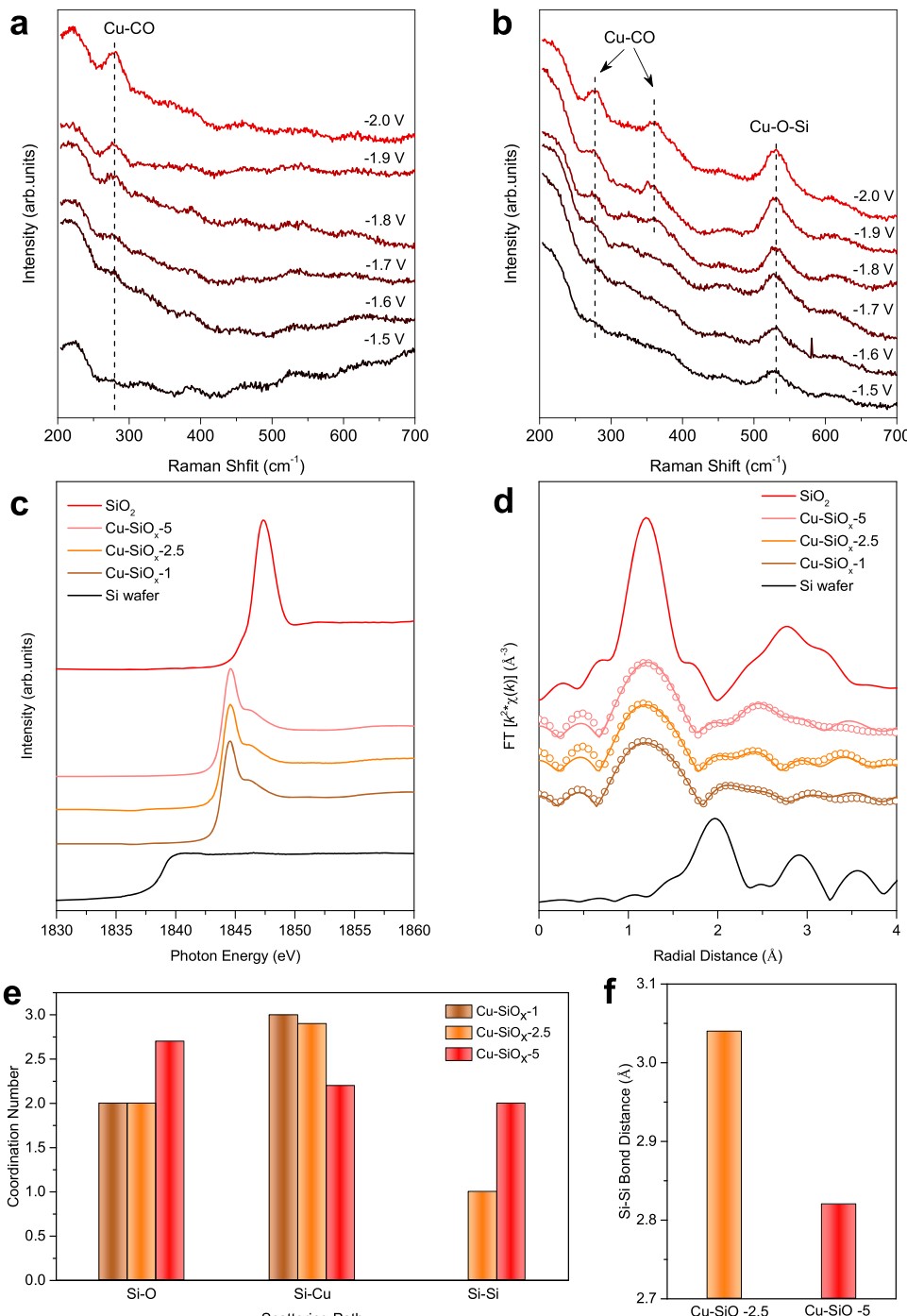

**Fig. 4 Spectroscopy studies of the Cu-SiO_x catalysts. a**, **b** In-situ Raman spectra of the bare Cu (**a**) and Cu-SiOx-2.5 (**b**) catalysts at a cell potential range of $-1.5 \sim -2.0$ V. Current density recorded at each condition is listed in Supplementary Table 4. **c** Ex-situ Si K-edge XANES spectra of different Cu-SiO_x catalysts and the Si standards (i.e., Si wafer and SiO_2). **d** The Si K-edge EXAFS spectra (solid lines) and corresponding fitting curves (open circles) of different Cu-SiO_x catalysts. The Si K-edge EXAFS spectra of Si wafer and SiO_2 standards are included for comparison. **e**, **f** Simulated coordination number (**e**) and Si–Si bond distance (**f**) of different Cu-SiO_x catalysts extracted from their Si K-edge EXAFS. In-situ Raman analysis was performed during CO_2RR, whereas ex-situ Si K-edge XAS was measured upon completion of CO_2RR at a full-cell potential of $-4.1$ V in 0.1 M KHCO_3.

(Supplementary Fig. 36d)[34]. The predominantly metallic Cu states of the Cu-SiO_x catalysts revealed by XAS further supports our view that silica clusters are deposited at the surface of Cu nanoparticles. This finding is also in line with the HAADF-STEM results (Supplementary Fig. 27).

The Si K-edge XANES of the Cu-SiO_x catalysts reveals an average +2 oxidation state of the Si species (Fig. 4c), with absorption onset energy ($1s \rightarrow 2p$ transition) and spectral characteristics analogous to the Si K-edge XANES of the molecular SiO deposited on highly oriented pyrolytic graphite, reported elsewhere[35]. Increasing the silica loading on Cu from 1 to 5% results in an enhancement of the Si K-edge white-line intensity and a blue shift of the absorption onset (Supplementary Fig. 37). A broadening of the white line indicates an overall Si p-charge depletion due to charge redistribution at the Cu-SiO_x interface with increasing silica loadings, and a positive energy

shift of Si 1$s$ absorption onset suggests a potential increase of the Si oxidation state from the charge redistribution process.

The Si K-edge EXAFS of the Cu-SiO$_x$ catalysts shows a strong peak at 1.3 Å, attributable to the Si–O bond in the first shell of Si atoms (Fig. 4d). We fit the Si K-edge EXAFS spectra of different Cu-SiO$_x$ catalysts with the relevant DFT models having similar silica loadings. We found that the Si local structures (i.e., coordination number and bond distance) extracted from the EXAFS fitting align well with those obtained from the DFT models (Fig. 4d, Supplementary Fig. 38 and Supplementary Table 5), indicating a general agreement between experimental and computational trends herein. From EXAFS fits, we also found a decrease in the Si-Cu coordination from 3 to 2 with the silica loading increasing from 2.5 to 5% (Fig. 4e), indicating that an increase in the silica loading to 5% diminishes the interaction between Cu and silica. Increasing the silica loading from 2.5 to 5% also results in the growth of silica molecules, evidenced by an increase in the Si–O coordination from 2 to 3, an increase in the Si–Si coordination from 1 to 2, and a ~7% contraction in the Si–Si bond from 3.0 to 2.8 Å (Fig. 4e, f).

Taken together, these results suggest that while Cu predominantly remains in a metallic state in the Cu-SiO$_x$ catalyst, the Si species preserve an oxidation state of ~+2 via Cu–O–Si interaction. An increase in the silica loading from 1 to 2.5% promotes the formation of stable Cu-SiO$_x$ interface sites with enhanced CO coverage to promote C–C coupling toward ethylene production, whereas a further increase in the silica loading to 5% diminishes the Cu and silica interaction due to the growth of silica molecules, destabilizing the Cu-SiO$_x$ interface and decreasing the CO content and therefore ethylene production on the electrode surface.

**System performance and catalyst stability.** To challenge the versatility of the Cu-SiO$_x$ catalyst, we assessed CO$_2$RR performance using dilute input gas streams with CO$_2$ concentrations in a range typical of flue gas[36]. We tested the CO$_2$RR performance of the Cu-SiO$_x$-2.5 catalyst across a cell potential window of −3.4 ~ −4.2 V by co-feeding CO$_2$ and N$_2$ gas feeds with the CO$_2$ concentrations ranging from 10 to 100%. We found that reducing the CO$_2$ concentration induces a slight decrease in current density (Supplementary Fig. 39), due to reduced CO$_2$ availability at the catalyst layer, evidenced by the increasing H$_2$ current densities at reduced CO$_2$ concentrations.

Regulating the CO$_2$ concentration in the full-cell potential window of −3.6 ~ −4.2 V results in a high FE$_{ethylene}$ of >50% with varied FE$_{CO}$ and FE$_{H_2}$ (Fig. 5a–c). At −3.6 V, we achieved a 52% FE$_{ethylene}$ (15% FE$_{CO}$) with 10% CO$_2$ feed, in contrast to a 37% FE$_{ethylene}$ with an increase of FE$_{CO}$ to 26% using 100% CO$_2$ at similar conditions. The sharp increase of FE$_{CO}$ at increasing CO$_2$ concentrations can be attributed to excessive local CO$_2$ in the catalyst layer, blocking active catalyst surface sites for CO dimerization toward ethylene formation and promoting instead CO production[17].

Increasing the potential to −4.2 V leads to a high FE$_{ethylene}$ of 64% (13% FE$_{H_2}$) with 100% CO$_2$ feed, whereas the FE$_{ethylene}$ decreases to 30% with a dramatic increase of FE$_{H_2}$ to 58% using 10% CO$_2$ at the same applied potential. A shift of product selectivity to H$_2$ at low CO$_2$ concentrations and at high negative potentials can be attributed to the CO$_2$ mass transport limitations (Fig. 5c and Supplementary Fig. 39). Taken together, these findings show that FE$_{ethylene}$ increases when we constrain CO$_2$ availability, which can be linked to modulation of the surface coverage of *CO$_2$, *CO, and *H at the catalyst layer[17]. As a result, we achieve a high FE$_{ethylene}$ from CO$_2$RR by tuning the local CO$_2$ concentration and applied potential/current density. Tuning

conditions for high FE$_{ethylene}$ and low cell potential, we achieve an 18% full-cell energy efficiency (EE) toward ethylene production across a broad range of CO$_2$ concentrations, 10–100% (Fig. 5d).

We also assessed the stability of the Cu-SiO$_x$-2.5 catalyst at different CO$_2$ concentrations to demonstrate stable selective ethylene electroproduction at industrial-scale reaction rates from various CO$_2$ feedstocks. We performed CO$_2$RR at 10, 40 and 100% CO$_2$ concentrations and achieved stable FE$_{ethylene}$ of >60% at 125, 250, and 300 mA cm$^{-2}$, respectively, over the course of continuous 55-h operation (Fig. 5e and Supplementary Fig. 40). We noted fluctuations in the first 5 h of stability testing with high CO$_2$ concentrations, a phenomenon we attribute to forming stable cathode:AEM:anode interfaces and stabilizing rates of gas, ion, and water transports with the MEA.

## Discussion

In this work, we show that introducing silica enables CO$_2$ activation and steers the reaction toward ethylene at the Cu-SiO$_x$ interface, at which silica might act as a dopant to promote Cu$^+$ sites and the formation of Cu–Cu$^+$ interface to assist C–C coupling[24,37]. However, in-situ Raman results show only two strong CO adsorption bands assigned to the metallic Cu–CO interactions. Neither any shifts of the Cu-CO Raman peaks nor additional Raman peaks related to the Cu$^+$-CO adsorption are observed on the Cu-SiO$_x$ catalyst. These findings suggest that the Cu$^+$ sites promoted by silica deposition does not contribute to the CO adsorption, and the CO adsorption energetics do not change with and without silica addition, consistent with our DFT calculations (Supplementary Figs. 8–12). We, therefore, conclude that silica, in this case, acts as a structural promoter to assist the electrochemical CO$_2$-to-ethylene conversion at the Cu-SiO$_x$ catalyst interface.

To demonstrate the universality of our oxide modulation strategy, we theoretically assessed the role of GeO$_x$—an analogous oxide to silica—on the C–C coupling reaction at the Cu-GeO$_x$ interface (Supplementary Figs. 41 and 42). By changing the GeO$_x$ loading ranging from 0 to 5%, we find a similar volcano-shaped dependence of the OCCOH* formation energy based on the GeO$_x$ loading. The Cu catalyst with a GeO$_x$ loading up to ~3% decreases the OCCOH* formation energy by ~−0.6 eV compared to the bare Cu catalyst, which we attributed to the formation of the Ge-O bond between O$^{\delta-}$ of the OCCOH* intermediate and GeO$_x$ at the Cu-GeO$_x$ interface.

In summary, we show an efficient oxide modulation strategy that modifies the activity and intermediate binding characteristics of Cu catalysts by creating active Cu-SiO$_x$ surface step sites. The MEA electrolyzer equipped with the Cu-SiO$_x$ catalyst enables FE$_{ethylene}$ of > 60%, current densities of up to 300 mA cm$^{-2}$, ethylene full-cell EEs of 18%, and continuous stable operation of 55 h with a wide range of input CO$_2$ concentrations (10–100%). With the aid of spectroscopic and electrochemical tools as well as DFT calculations, we reveal that the silica addition can decrease the formation energies of OCOH* and OCCOH* by forming strong Si–O or Si–C bonds at a stable Cu-SiO$_x$ step site, and thereby accelerate ethylene production. The improved reaction energetics achieved in the oxide-promoted Cu catalyst would motivate new catalyst design strategies for CO$_2$RR.

## Methods

**DFT calculations.** Density functional theory calculations were carried out with the Vienna Ab Initio Simulation Package (VASP) code[38,39]. Perdew–Burke–Ernzerhof (PBE) functionals[40] were used to treat the exchange-correlation interactions, and the projector-augmented wave (PAW) method[41] was used to solve the ion-electron interactions in the periodic system. All the configurations were first optimized under vacuum calculations and then converged under implicit solvation conditions. Details of DFT calculations were included in Supplementary Note 1.

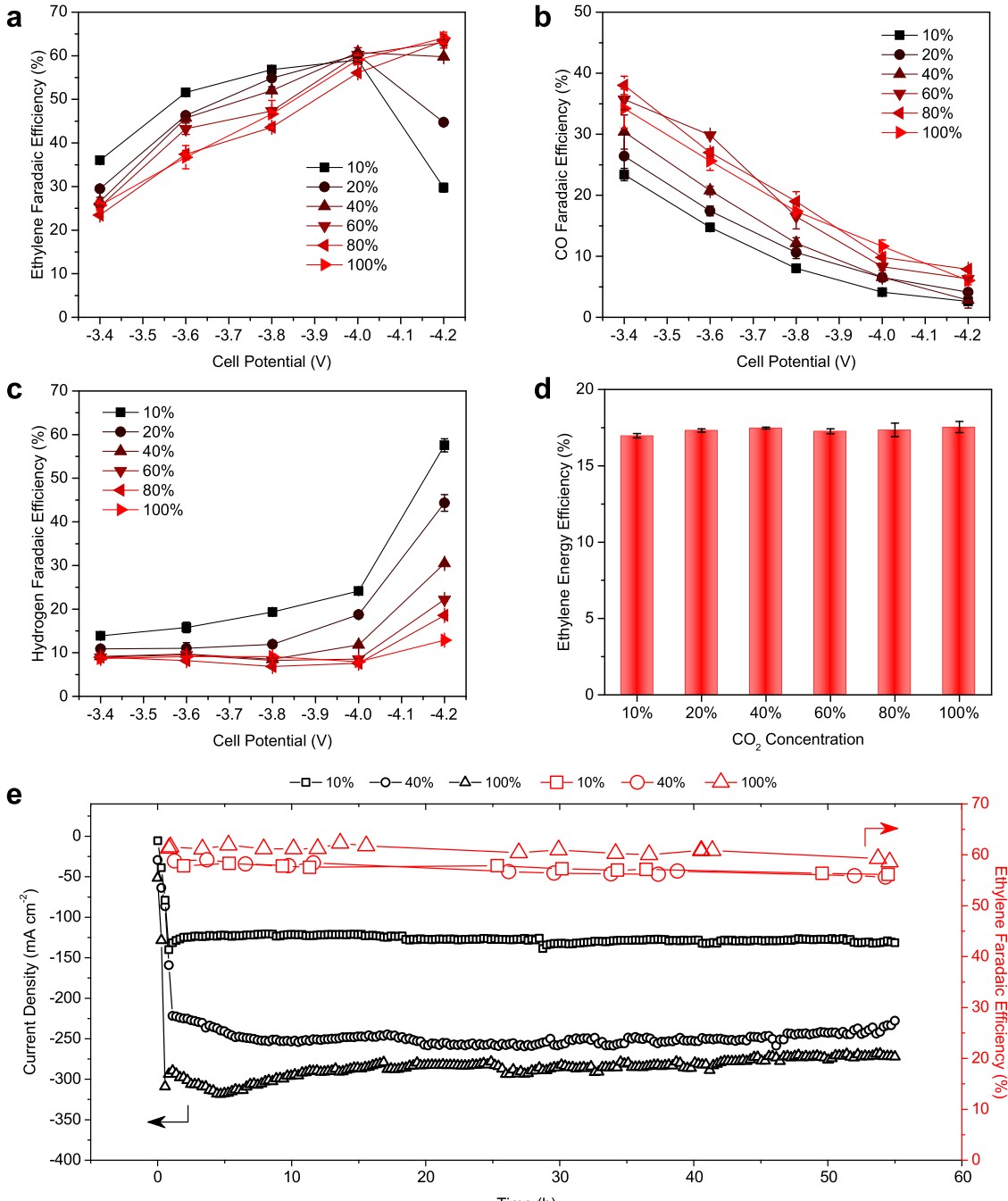

**Fig. 5 Applied electrochemical performance of 2.5% silica-loaded Cu catalyst in a MEA electrolyzer with 0.1 M KHCO₃ anolyte. a–d** Effect of CO₂ concentration on FE$_{ethylene}$ (**a**), FE$_{CO}$ (**b**), FE$_{hydrogen}$ (**c**), and EE$_{ethylene}$ (**d**). Error bars are means ± SD ($n = 3$ replicates). **e** Extended operational stability of the MEA equipped with the Cu-SiO$_x$ under CO₂ concentration of 10% at −3.8 V, 40% at −4.0 V, and 100% at −4.1 V.

**Catalysts preparation**. For the synthesis of 2.5% silica-loaded Cu catalysts, 0.5984 g of CuCl₂.2H₂O (Sigma-Aldrich, ACS reagent, ≥99.0%) was dissolved into 180 mL ethanol under magnetic stirring. Then 90 μL of silicon tetrachloride (Sigma-Aldrich, 99.998% trace metals basis) was injected into the Cu(II) solutions. Afterward, 0.6 g of sodium borohydride (Sigma-Aldrich, powder, ≥98.0%) dispersed in 120 mL of ethanol was added into the above-mentioned solution under stirring. The color of the solutions changed from blue to darkish. After stirring for 5 min, precipitates formed in the solution were kept for 30 mins and then centrifuged. The as-prepared Cu-SiO$_x$ precipitates were washed with ethanol under intense ultrasonication at least three times to remove all unreacted reagents. The Cu-SiO$_x$ catalysts were then dried under vacuum at room temperature overnight. The silica-loaded Cu catalysts with different Si/Cu ratios of 0, 0.01, and 0.05 were synthesized by following a procedure similar to that described above, except for changing the amount of silicon tetrachloride.

**Materials characterization**. A Hitachi S-5200 SEM/STEM operating at 1 kV was employed to analyze the surface morphology of catalysts. TEM analysis was performed using a Hitachi HF-3300 instrument with an acceleration voltage of 300 kV. An aberration-corrected FEI Titan 80–300 equipped with a CEOS probe and image corrector was used for high-resolution TEM images and EDX. The microscope was operated at an acceleration voltage of 200 kV and STEM micrographs were captured using a HAADF detector (Fischione). The microscope was equipped with four-quadrant windowless silicon drift detectors EDX (Super-X) which increased the angle of collection, and hence, collection efficiency. During EDX mapping, the electron beam used for STEM imaging was rastered over the particles and the characteristic X-rays (resulting from electron-matter interaction) were collected by EDX detectors (four of them available in this microscope). For each EDX map, the area was exposed to the electron beam for 10–15 min, depending on the number of counts collected and the qualitative signal-to-noise ratio. In general,

once the overall number of counts (detected X-rays) reached ~0.5 million (depending somewhat on the area size and elements), the maps and the elemental signals were clear. The Agilent 5110 ICP-OES system was employed to determine the elemental composition. P-XRD was performed on a MiniFlex600 instrument with a copper target ($\lambda = 1.54056$ Å) at room temperature. XPS measurements were carried out on a K-Alpha XPS spectrometer (PHI 5700 ESCA System), using Al Kα X-ray radiation (1486.6 eV) for excitation. In-situ XAS measurements at the Cu K-edge were carried out at the 9BM beamline of Advanced Photon Source (APS, Argonne National Laboratory, Lemont, Illinois, USA) using a home-made flow cell[42]. Ex-situ Si K-edge experiments were conducted at the Soft X-ray Microcharacterization Beamline (SXRMB) of the Canadian Light Source (CLS, Saskatoon, Saskatchewan, Canada) by sealing samples in Kapton tape after the $CO_2RR$ testing and vacuum drying. Ex-situ XAS measurements at the Cu K-edge were also recorded at the SXRMB beamline together with ex-situ Si K-edge testing to exclude material oxidation from ex-situ measurements (Supplementary Fig. 43). Fluorescence yield was recorded using silicon drift detectors at both the 9BM and SXRMB beamlines. XANES data were analyzed using Athena software included in a standardized IFEFFIT package[43]. EXAFS fittings were performed using the FEFF software included in the IFEFFIT package[44]. In-situ Raman experiments were performed using a Renishaw inVia Raman microscope in a home-made flow cell (Supplementary Fig. 44) with a water immersion objective. During testing, $CO_2$ with a constant flow rate of 20 s.c.c.m. flowed through the gas compartment, whereas 0.1 M KHCO$_3$ solution with a flow rate of 0.2 mL min$^{-1}$ flowed through the electrolyte compartment. A Pt wire was used as the counter electrode, and a 785-nm laser was used for Raman testing. Signals were recorded using a 5-s integration and averaging ten scans.

**Electrode preparation.** The cathodes were manufactured via a two-step fabrication procedure. The first step was evaporating Cu (Kurt J. Lasker Company) on a polytetrafluoroethylene (PTFE) substrate at a constant sputtering rate of 0.45 A s$^{-1}$ at $10^{-6}$ Torr until the sputtering thickness of 150 nm was achieved. The second step was spray-depositing a homogeneous ink of Cu catalysts, polymeric binder (Aquivion, D79-25BS), and methanol until the catalyst loading of 1.25 mg cm$^{-2}$ was achieved. The anodes for the OER were manufactured via a five-step fabrication procedure. The procedure involved (i) etching the titanium (Ti) mesh (Fuel Cell Store) in 5 M HCl at 70 °C for 30 min, (ii) preparing a catalyst ink of iridium (III) oxide dehydrate (Alfa Aesar, 99.99%), 2-propanol and HCl, (iii) immersing the etched Ti mesh into the catalysts ink for 30 s, (iv) drying and sintering the resulting Ti meshes for 15 min at 100 and 500 °C, respectively, and (v) repeating the immersion, drying, and sintering steps until the total IrO$_2$ loading of 1.5 mg cm$^{-2}$ was achieved.

**CO$_2$ electroreduction.** The $CO_2RR$ performance assessments of the cathode Cu-SiO$_x$ catalysts were made via a custom-made electrochemical test station. The station was equipped with a potentiostat and booster, $CO_2RR$ electrolyzer (a membrane electrode assembly electrolyzer purchased from Dioxide Materials), mass flow controller, humidifier, and peristaltic pump.

For $CO_2RR$ testing, the electrolyzer consisted of anode and cathode flow-field plates with a geometric flow area of 5 cm$^2$. The anode and cathode flow-field plates were made of titanium and stainless steel, respectively, and they were mainly responsible for the distribution of anolyte (0.1 M KHCO$_3$) and reactant ($CO_2$) over the surface of the respective electrodes. The anode and cathode electrodes were mounted on their respective flow-field plates and physically isolated from each other by an AEM (Sustainion X37-50 Grade RT). The AEM was activated for at least 24 h in 1 M KOH and rinsed with deionized (DI) water continuously for 15 min to remove the KOH from the surface before the experiments. Upon completion of the electrolyzer assembly, the anolyte (0.1 M KHCO$_3$) and humidified $CO_2$ were supplied to their respective compartments with constant flow rates of 10 mL min$^{-1}$ and 80 s.c.c.m., respectively. The $CO_2RR$ was then initiated and the corresponding current was recorded. Potentials reported in MEA electrolyzer were non-iR corrected full-cell potentials.

Methods used for quantifying the $CO_2RR$ products were reported elsewhere[45]. Briefly, a gas chromatography (GC, PerkinElmer Clarus 680), equipped with a Molecular Sieve 5 A capillary column and a packed Carboxen-1000 column, was used to analyze the $CO_2RR$ gas products. While hydrogen was quantified using a thermal conductivity detector, ethylene was measured using a flame ionization detector, and argon (Linde, 99.999%) was used as a carrier gas. One-dimensional $^1$H Nuclear magnetic resonance spectroscopy ($^1$H-NMR) coupled with an Agilent DD2 500 spectrometer, was used to evaluate liquid products with Dimethyl sulfoxide (DMSO) diluted in D$_2$O as the internal standard. Calculation details of the FEs toward gas and liquid products of $CO_2RR$ as well as those of the ethylene full-cell EE can be found in Supplementary Note 2. The FE and FE-derived data were obtained by averaging the FEs obtained from three independent experiments, and the data were presented with error bars.

## Data availability

The data that support the findings of this study are available from the corresponding author on reasonable request.

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

## Acknowledgements

This work has received financial support from the Ontario Research Fund Research-Excellence Program, the Natural Sciences and Engineering Research Council (NSERC) of Canada, the CIFAR Bio-Inspired Solar Energy Program, and the University of Toronto Connaught grant. This research used synchrotron resources of the Advanced Photon Source (APS), an Office of Science User Facility operated for the U.S. Department of Energy (DOE) Office of Science by Argonne National Laboratory, and was supported by the U.S. DOE under Contract No. DE-AC02-06CH11357, and the Canadian Light Source and its funding partners. This research also used infrastructure provided by the Canada Foundation for Innovation and the Ontario Research Fund. We thank Dr. T.P. Wu, Dr. Y.Z. Finfrock and Dr. L. Ma for technical support at 9BM beamline of APS. D.S. acknowledges the NSERC E.W.R Steacie Memorial Fellowship. J.L. acknowledges the Banting Postdoctoral Fellowships program. DFT calculations were performed on the Massachusetts Green High Performance Computing Center (MGHPCC). The authors also acknowledge the Texas Advanced Computing Center (TACC) at the University of Texas at Austin for partially providing HPC resources that have contributed to the research results reported within this paper. Our thanks also goes to institutional faculty start-up funds from University of Massachusetts Lowell.

## Author contributions

E.H.S. and D.S. supervised the project. J.L. and A.O. conceived the idea, carried out all the experiments, and wrote the manuscript. F.C. supervised and M.W. performed the DFT simulations and both wrote the corresponding sections. Z.Y.W. assisted the DFT discussion. J.L. and Y.F.H. performed and analyzed synchrotron data. A.O. and J.L. performed the in-situ Raman measurements. Y.H.W., R.Z., D.R., and B.C. assisted in SEM and TEM analysis. D.-H.N. and J.W. conducted XRD and XPS measurements. F.W.L., Y.X., X.W., M.C.L., and M.G. assisted in electrochemical experiments and analysis. All authors discussed the results and assisted during manuscript preparation.

## Competing interests

The authors declare no competing interests.
