## [Peer Review File · Nature Communications]

Reviewer #1 (Remarks to the Author):

The paper by Li et al. entitled "Silica-copper catalyst interfaces enable C-C coupling toward ethylene electro-synthesis" presents interesting catalytic data on the electrochemical reduction of Cu systems containing Si²⁺.

The reactivity data are interesting, presented with high scientific rigor, but not exceptional. However the initial hypothesis of creating active surface step sites on Cu that assist CO₂ activation and its ensuing reduction toward ethylene is only explored by a, certainly extensive, computational study. One would expect a more comprehensive study with complementary experimental data aimed to identify the role (geometric and electronic) of Si cations and how these evolve in situ together with Cu.

Therefore I am afraid that this study seems still premature or else I could not recommend for publication in Nature Communication at the actual state. A more specialized journal could be suitable.

Reviewer #2 (Remarks to the Author):

This work deals with a silica doping way to enrich active surface step sites for enhanced electroreduction of CO₂ to ethylene. The authors first conducted DFT calculations, and predicted the Cu₀-Si²⁺ step sites would be active by lowering the formation energies of key intermediates of OCOH* and OCCOH* for ethylene formation. Then a series of model silica doped Cu-SiO_x catalysts were developed through a one-pot coprecipitation method, and evaluated in an integrated MEA electrolyzer. The improvement compared with bare Cu catalysts is clear, this is evidenced by the higher FE and current density towards ethylene. However, the synthesis and understanding of model catalysts as well as the interpretation on structure-performance relationship is rather mediocre, even coarse. In my view, it would be proper for a resubmission with the relevant issues addressed.

1. I would rather the nature of the Cu-SiO_x interface. The author proposed the Cu₀-Si²⁺ step sites. Does this mean the Si-species physically adsorb on the Cu plane? The lack of the bridging interactions, electrostatic attraction as well as the absence of confined effects would hardly ensure an effective catalytic effect. Thus, there is sustainable work to be done to clarify its boosting effects.
2. The authors stated that when the Si loading is increased to 4.7%, the energetically favorable silica dopant forms a Si-O-Si layer and fails to form Si-C or Si-O, however, a Si loading of 3.1% is rather effective. The question is what is the status of the Si-species at 3.1% and 4.7 %, clusters? NPs?
3. Evidenced by the CO₂RR performances, creating the new Cu-Si sites enhanced the activity and product selectivity. The new sites render the formation of the key intermediates more easily with lower activation energy. However, the creating of the step sites would also probably just improve the accessibility of the transported CO₂. The apparent activation energy of bare Cu as well as the modified catalysts would give more direct proof.

Reviewer #3 (Remarks to the Author):

In this manuscript, the authors report an oxide doping strategy based on silica to create abundant Cu-SiO_x interface stepped sites, assisting CO₂ activation and the formation of the OCCOH* intermediate. The result is interesting. This strategy provides a new possibility for enhancing the electrochemical activity and selectivity for CO₂ reduction to ethylene. However, there are still some important issues that should be well addressed before publication.

Comment 1: The results show that a Cu-SiO_x catalyst with a Si loading of ~3% increases the efficiency of CO₂-to-ethylene conversion. The reason for this result should be discussed. I think excess SiO_x is unfavorable, which would reduce the accessibility of CO₂ to the active site. It is very important to know the coverage ratio of SiO_x on Cu.

Comment 2: The authors declare a uniform distribution of Si in Cu, as evidenced by elemental mapping. However, the morphology of the Cu-SiO_x catalyst is unclear, especially the assembly style of Cu and SiO_x. The size and shape of SiO_x may be key to the performance of the catalyst. The catalyst should be characterized by more techniques.

Comment 3: It is better to give the in-situ characterization for the local atomistic structure of the material during electrocatalysis.

Comment 4: The results in Fig. 4a-c should be discussed in more details. An explanation for the FE changes of different products is needed.

Reviewer 1

The paper by Li et al. entitled “Silica-copper catalyst interfaces enable C-C coupling toward ethylene electro-synthesis” presents interesting catalytic data on the electrochemical reduction of Cu systems containing Si²⁺. The reactivity data are interesting, presented with high scientific rigor, but not exceptional. However the initial hypothesis of creating active surface step sites on Cu that assist CO₂ activation and its ensuing reduction toward ethylene is only explored by a, certainly extensive, computational study. One would expect a more comprehensive study with complementary experimental data aimed to identify the role (geometric and electronic) of Si cations and how these evolve in situ together with Cu. Therefore I am afraid that this study seems still premature or else I could not recommend for publication in Nature Communication at the actual state. A more specialized journal could be suitable.

In light of the Reviewer’s comment, we have performed additional microscopy measurements on the 2.5% silica-loaded Cu catalyst to characterize the *geometry* of Si cations on Cu. We found that the Si species are evenly distributed at the Cu catalyst surface, and the structure of Si species remains intact with an oxidation state of +2 both before and after CO₂RR (**Figure R1 – R4**), consistent with DFT calculations (**Fig. 1a**).

We have included these results in the revised manuscript (**Page 7, Line 150**):

“Transmission electron microscopy and scanning electron microscopy images show nanoparticles in an aggregated form and a mean size of ~50 nm. The Cu morphology is unchanged with silica loading (**Fig. 2a** and **Supplementary Fig. 20**). The catalysts with/without silica loading lead to similar Cu patterns, seen in X-ray powder diffraction (XRD) measurements (**Supplementary Fig. 21**). We examined and compared the structure of the Cu-SiO_x catalyst before and after CO₂RR using high-angle annular dark-field scanning transmission electron microscopy (HAADF-STEM) and energy-dispersive X-ray (EDX) mapping. We found that the Si species are evenly distributed at the Cu catalyst surface, and that the structure of Si species remains intact before and after the reaction, showing an oxidation state of +2 revealed by X-ray photoelectron spectroscopy (XPS, **Fig. 2b – f** and **Supplementary Fig. 22 – 25**). Using HAADF-STEM and XPS, we also characterized the Cu-SiO_x catalysts with silica loadings of 2.5% (**Fig. 2**) and 5% (**Supplementary Fig. 26**), and found similar silica structures in both samples and detected no aggregation of silica, suggesting that silica clusters are atomically distributed on the Cu surface.”

To reveal the *electronic* interaction between Si and Cu, we have conducted X-ray absorption measurements at the Cu K-edge and the Si K-edge (**Figure R5 – R9** and **Table R1**), on various Cu-SiO_x-M catalysts (M = 1, 2.5 or 5 which represents the silica loading percentage on Cu).

Using hard X-ray, in-situ Cu K-edge XAS spectra of different Cu-SiO_x catalysts were recorded at the 9BM beamline of the Advanced Photon Source (9BM@APS, <https://www.aps.anl.gov/Spectroscopy/Beamlines/9-BM>). The results show that the Cu species remain the metallic state (**Figure R5a** and **R6**).

Since the absorption energy of Si K-edge (~1839 eV) is below the energy range of 9BM beamline (2.1 – 24 keV) and the soft X-ray absorption measurement requires ultrahigh vacuum system, we conducted the Si K-edge experiments at the Soft X-ray Microcharacterization Beamline of the Canadian Light Source (SXRMB@CLS, 1.7 – 10 keV, https://www.lightsource.ca/beamlines/details/soft_xray_microcharacterization_beamline_sxrmb.html). However, the sample testing with a mail-in service was the only option at CLS due to a partial lockdown at the moment. Due to this circumstance, we instead performed ex-situ Si K-edge testing. Samples after CO₂RR were rinsed with DI water, dried using N₂ gas and then stored under vacuum at room temperature overnight, and sealed with Kapton tape in N₂-filled glove box. Ex-situ Si K-edge measurements show that the Si species in different Cu-SiO_x catalysts preserve a ~+2 oxidation state after CO₂RR and no oxidation of Cu is observed from synchronous ex-situ Cu K-edge XAS testing at SXRMB beamline (**Figure R7**), suggesting the nature of Cu-SiO_x interface sites for CO₂RR.

We also showed that the Si oxidation slightly increases with increasing the silica loading from 1% to 5% (**Figure R5b** and **Figure R8**). We then simulated the Si K-edge EXAFS spectra by applying the DFT models used for energy calculations in **Fig. 1** (**Figure R5c – g**, **Figure R9** and **Table R1**). We showed consistent Si local structures (i.e. coordination number and bond distance) extracted from the DFT modeling and the EXAFS fitting.

We have added this discussion to the revised manuscript (**Page 8, Line 164**):

“To investigate the electronic interaction between silica and Cu, we performed X-ray absorption spectroscopy (XAS) measurements at the Cu K-edge and Si K-edge. In-situ X-ray absorption near-edge structure (XANES) at the Cu K-edge reveals the metallic nature of Cu active sites (**Fig. 3a**), corroborated by the extended X-ray absorption fine structure (EXAFS, **Supplementary Fig. 27**). XANES at the Si K-edge reveals an average +2 oxidation state of the Si species based on the dipole-allowed (1s → 2p transition) absorption onset energy position (**Fig. 3b**). Increasing the silica loading on Cu from 1% to 5% results in an enhancement of the Si K-edge white-line intensity and a blue shift of the absorption onset (**Supplementary Fig. 28**), indicating that the Si oxidation state increases with increasing silica loading, in line with our DFT predictions (**Fig. 1a**).

The Si K-edge EXAFS shows a strong peak at 1.3 Å, attributable to the Si-O bond in the first shell of Si atoms (**Supplementary Fig. 29**). We fit the Si K-edge EXAFS spectra of different Cu-SiO_x catalysts with the relevant DFT models having the similar silica loadings. We found that the Si local structures (i.e. coordination number and bond distance) extracted from the EXAFS fitting align well with those obtained from the DFT modeling (**Fig. 3c – e, Supplementary Fig. 30 and Supplementary Table 3**), indicating general agreement between experimental and computational trends herein. From EXAFS fits, we also found a decrease in the Si-Cu coordination from 3 to 2 with the silica loading increasing from 2.5% to 5% (**Fig. 3f**), indicating that an increase in the silica loading to 5% diminishes the interaction between Si and Cu. Increasing the silica loading from 2.5% to 5% also results in a stronger Si-O-Si layer, evidenced by an increase in the Si-O coordination from 2 to 3, an increase in the Si-Si coordination from 1 to 2, and a ~7% contraction in the Si-Si bond from 3.0 Å to 2.8 Å (**Fig. 3f and 3g**).

Taken together, these results suggest that while Cu remains in a metallic state in the Cu-SiO_x catalyst, the Si species preserves an oxidation state of +2 at the Cu surface. An increase of the silica loading from 1% to 2.5% promotes the formation of stable Cu⁰-Si²⁺ interface sites, whereas a further increase in the Si loading to 5% diminishes the Si and Cu interaction and forms a strong Si-O-Si layer, destabilizing the Cu-SiO_x interface.”

Figure R1. Characterization of 2.5% silica-loaded Cu catalyst before CO₂RR. a, b TEM image (a) and HAADF-STEM image (b). **c** EDX elemental mapping of Si. **d** XPS spectra of the Si 2p. Results show that Si species are homogeneous distributed at the Cu catalyst and show an oxidation state of +2.

Figure R2. Characterization of 2.5% silica-loaded Cu catalyst after CO₂RR. a, b TEM image (a) and HAADF-STEM image (b). c, d EDX elemental mapping of Cu (c) and Si (d). e EDX spectrum of the Cu-SiO_x catalyst collected from the area of (b). f, g XPS spectra of the Cu 2p (f) and Si 2p (g). Results show that Si species are homogeneous distributed on the Cu catalyst and maintain an oxidation state of +2 after CO₂RR.

Figure R3. Characterization of 2.5% silica-loaded Cu catalyst after CO₂RR. a HAADF-STEM image. **b, c** EDX spectra of the Cu-SiO_x catalyst collected from the surface Area #1 and bulk Area #2 indicated in (a), suggesting that Si species are distributed on the Cu surface rather than in the Cu bulk.

Figure R4. Characterization of 2.5% silica-loaded Cu catalyst after CO₂RR. a, b HAADF-STEM images at different length scales. c, d High-resolution electron energy loss spectra of Cu (c) and Si (d) collected from the area of (b), showing that no significant aggregation of Si species is observed.

Figure R5. X-ray absorption characterization of various $\text{Cu-SiO}_x\text{-M}$ catalysts ($M = 1, 2.5$ or 5 , corresponding to the silica loading percentage in Cu). **a** Operando Cu K-edge XAS spectra of different Cu-SiO_x catalysts and the Cu K-edge XAS spectra of Cu foil and Cu_2O standards. **b** Ex-situ Si K-edge XAS spectra of different Cu-SiO_x catalysts and the Si standards (i.e. Si wafer and SiO_2). **c-e** Si K-edge EXAFS spectra (open circles) and corresponding fitting spectra (red curve) of various Cu-SiO_x catalysts at a silica loading of 1% (**c**), 2.5% (**d**) and 5% (**e**). Insets show the relevant DFT models applied for EXAFS fittings, at a theoretical silica

loading of 1.6% in (c), 3.1% in (d) and 4.7% in (e). **f, g** Simulated coordination number (**f**) and Si-Si bond distance (**g**) of different Cu-SiO_x catalysts extracted from their Si K-edge EXAFS.

Figure R6. Operando Cu K-edge EXAFS spectra of different Cu-SiO_x catalysts and the Cu K-edge EXAFS spectra of Cu foil and Cu₂O standards.

Figure R7. Ex-situ XAS characterization of the Cu-SiO_x catalyst at a silica loading of 2.5%. a Si K-edge XANES of the Cu-SiO_x catalyst, Si wafer and SiO₂. **b, c** Cu K-edge XANES (b) and EXAFS (c) of the Cu-SiO_x catalyst, Cu foil and Cu₂O.

Figure R8. Si K-edge XANES spectra of various Cu-SiO_x catalysts at silica loadings in the range of 1% and 5%. An enhancement in the white-line intensity indicates the increase in the Si oxidation state with increasing silica loading on the Cu catalyst.

Figure R9. EXAFS fitting of the Si K-edge. a-c The Si K-edge EXAFS spectra (open circles) and corresponding scattering paths from EXAFS fitting (solid lines) of various Cu-SiO_x catalysts at a silica loading of 1% (a), 2.5% (b) and 5% (c).

Table R1. Comparison of the structures of various Cu-SiO_x catalysts through DFT modeling (insets of **Figure R5 c-e**) and fitting of EXAFS experimental data (**Figure R5 c-e**) at the Si K-edge (CN: coordination number; R: bond distance).

Sample	Shell	CN _{DFT}	R _{DFT} (Å)	CN _{EXAFS}	R _{EXAFS} (Å)
Cu-SiO _x -1	Si-O	2	1.62	2	1.61
	Si-Cu ₁	1	2.30	1	2.26
	Si-Cu ₂	1	2.45	1	2.46
	Si-Cu ₃	1	2.57	1	2.67
Cu-SiO _x -2.5	Si-O	2	1.64	2	1.61
	Si-Cu ₁	1	2.32	0.8	2.31
	Si-Cu ₂	1	2.42	0.8	2.53
	Si-Cu ₃	1	2.63	1.3	2.80
	Si-Si	1	3.0	1	3.04
Cu-SiO _x -5	Si-O ₁	1	1.64	1	1.74
	Si-O ₂	2	1.68	1.7	1.61
	Si-Cu ₁	1	2.40	0.8	2.33
	Si-Cu ₂	1	2.72	0.7	2.63
	Si-Cu ₃	1	2.78	0.7	2.69
	Si-Si	2	2.99	2	2.82

Reviewer #2

This work deals with a silica doping way to enrich active surface step sites for enhanced electroreduction of CO₂ to ethylene. The authors first conducted DFT calculations, and predicted the Cu⁰-Si²⁺ step sites would be active by lowering the formation energies of key intermediates of OCOH and OCCOH* for ethylene formation. Then a series of model silica doped Cu-SiO_x catalysts were developed through a one-pot coprecipitation method, and evaluated in an integrated MEA electrolyzer. The improvement compared with bare Cu catalysts is clear, this is evidenced by the higher FE and current density towards ethylene. However, the synthesis and understanding of model catalysts as well as the interpretation on structure-performance relationship is rather mediocre, even coarse. In my view, it would be proper for a resubmission with the relevant issues addressed.*

We thank Reviewer #2 and detail below our actions taken in order to explore in greater depth the structure-performance relationship.

1. I would rather the nature of the Cu-SiO_x interface. The author proposed the Cu⁰-Si²⁺ step sites. Does this mean the Si-species physically adsorb on the Cu plane? The lack of the bridging interactions, electrostatic attraction as well as the absence of confined effects would hardly ensure an effective catalytic effect. Thus, there is sustainable work to be done to clarify its boosting effects.

We explored the electronic interaction between silica and Cu and the interface using X-ray absorption spectroscopy.

The Cu K-edge XAS spectra, both at the near-edge (XANES) and extended edge (EXAFS) regions, of different Cu-SiO_x catalysts show that the Cu remains in the metallic state during CO₂RR (**Figure R10**).

A detailed electronic interaction between silica and Cu comes from the Si K-edge. XANES spectra of different Cu-SiO_x catalysts show an absorption onset of ~1844 eV (**Figure R11a**), indicating that the Si species has a +2 oxidation state. The Si oxidation increases with increasing silica loading (**Figure R11b**). Fittings of the EXAFS spectra of various Cu-SiO_x catalysts (**Figure R12**) show that an increase in the silica loading from 1% to 2.5% promotes stable Cu⁰-Si²⁺ step sites, whereas a further increase in the silica loading to 5% diminishes the Si and Cu interaction with reduced Si-Cu coordination, destabilizing the Cu-SiO_x interface with reduced capability to catalyze CO₂, in line with our DFT calculations.

We now have clarified the Cu-SiO_x electronic interaction in the revised manuscript (**Page 8, Line 164**):

“To investigate the electronic interaction between silica and Cu, we performed X-ray absorption spectroscopy (XAS) measurements at the Cu K-edge and Si K-edge. In-situ

X-ray absorption near-edge structure (XANES) at the Cu K-edge reveals the metallic nature of Cu active sites (**Fig. 3a**), corroborated by the extended X-ray absorption fine structure (EXAFS, **Supplementary Fig. 27**). XANES at the Si K-edge reveals an average +2 oxidation state of the Si species based on the dipole-allowed ($1s \rightarrow 2p$ transition) absorption onset energy position (**Fig. 3b**). Increasing the silica loading on Cu from 1% to 5% results in an enhancement of the Si K-edge whiteline intensity and a blue shift of the absorption onset (**Supplementary Fig. 28**), indicating that the Si oxidation state increases with increasing silica loading, in line with our DFT predictions (**Fig. 1a**).

The Si K-edge EXAFS shows a strong peak at 1.3 Å, attributable to the Si-O bond in the first shell of Si atoms (**Supplementary Fig. 29**). We fit the Si K-edge EXAFS spectra of different Cu-SiO_x catalysts with the relevant DFT models having the similar silica loadings. We found that the Si local structures (i.e. coordination number and bond distance) extracted from the EXAFS fitting align well with those obtained from the DFT modeling (**Fig. 3c – e**, **Supplementary Fig. 30** and **Supplementary Table 3**), indicating general agreement between experimental and computational trends herein. From EXAFS fits, we also found a decrease in the Si-Cu coordination from 3 to 2 with the silica loading increasing from 2.5% to 5% (**Fig. 3f**), indicating that an increase in the silica loading to 5% diminishes the interaction between Si and Cu. Increasing the silica loading from 2.5% to 5% also results in a stronger Si-O-Si layer, evidenced by an increase in the Si-O coordination from 2 to 3, an increase in the Si-Si coordination from 1 to 2, and a ~7% contraction in the Si-Si bond from 3.0 Å to 2.8 Å (**Fig. 3f** and **3g**).

Taken together, these results suggest that while Cu remains in a metallic state in the Cu-SiO_x catalyst, the Si species preserves an oxidation state of +2 at the Cu surface. An increase in the silica loading from 1% to 2.5% promotes the formation of stable Cu⁰-Si²⁺ interface sites, whereas a further increase in the Si loading to 5% diminishes the Si and Cu interaction and forms a strong Si-O-Si layer, destabilizing the Cu-SiO_x interface.”

Figure R10. In-situ Cu K-edge XAS spectra of Cu-SiO_x catalysts. a, b The Cu K-edge XANES (a) and EXAFS (b) of the Cu standards (i.e. Cu foil and Cu₂O) and different Cu-SiO_x catalysts with silica loadings in the range of 1% and 5%.

Figure R11. Si K-edge XAS spectra of Cu-SiO_x catalysts. a The Si K-edge XANES spectra of the Si standards (i.e. Si wafer and SiO₂) and different Cu-SiO_x catalysts with silica loadings in the range of 1% and 5%. **b** An overlay of the Si K-edge XANES spectra of different Cu-SiO_x catalysts.

Figure R12. Si K-edge EXAFS fitting of Cu-SiO_x catalysts. a-c The Si K-edge EXAFS

spectra (open circles) and corresponding scattering paths from EXAFS fitting (solid lines) of various Cu-SiO_x catalysts at a silica loading of 1% (a), 2.5% (b) and 5% (c). Structural models indicated below the EXAFS spectra show the relevant DFT models applied for EXAFS fittings, at a theoretical silica loading of 1.6% in (a), 3.1% in (b) and 4.7% in (c). d, e Simulated coordination number (d) and Si-Si bond distance (e) of different Cu-SiO_x catalysts extracted from their Si K-edge EXAFS.

2. The authors stated that when the Si loading is increased to 4.7%, the energetically favorable silica dopant forms a Si-O-Si layer and fails to form Si-C or Si-O, however, a Si loading of 3.1% is rather effective. The question is what is the status of the Si-species at 3.1% and 4.7 %, clusters? NPs?

Using HAADF-STEM mapping and XPS, we have now characterized and compared the structure of Cu-SiO_x-2.5 (Figure R13) and Cu-SiO_x-5 (Figure R14) catalysts with silica loading 2.5% and 5%. We found that the structure of the Si species in the two samples are similar, and we observe no aggregation of SiO_x in the form of NPs, suggesting that the Si species are atomically distributed on the Cu surface for the silica loadings between 0% and 5%.

We further explored the structural differences between the Si species in different Cu-SiO_x catalysts at the atomic level using the Si K-edge EXAFS. We detail in our reply to comment 1 of the Reviewer that too much silica (e.g. a silica loading of 5%) diminishes the Cu-Si interaction and leads to a strong Si-O-Si layer, destabilizing the Cu-SiO_x interface for C-C coupling – a key reaction step toward ethylene production.

We have included these discussions in the revised manuscript (Page 7, Line 150):

“Transmission electron microscopy and scanning electron microscopy images show nanoparticles in an aggregated form and a mean size of ~50 nm. The Cu morphology is unchanged with silica loading (Fig. 2a and Supplementary Fig. 20). The catalysts with/without silica loading lead to similar Cu patterns, seen in X-ray powder diffraction (XRD) measurements (Supplementary Fig. 21). We examined and compared the structure of the Cu-SiO_x catalyst before and after CO₂RR using high-angle annular dark-field scanning transmission electron microscopy (HAADF-STEM) and energy-dispersive X-ray (EDX) mapping. We found that the Si species are evenly distributed at the Cu catalyst surface, and that the structure of Si species remains intact before and after the reaction, showing an oxidation state of +2 as revealed by X-ray photoelectron spectroscopy (XPS, Fig. 2b – f and Supplementary Fig. 22 – 25). Using HAADF-STEM and XPS, we also characterized the Cu-SiO_x catalysts with the silica loadings of 2.5% (Fig. 2) and 5% (Supplementary Fig. 26), and found similar silica structures in both samples and detected no aggregation of silica, suggesting that silica clusters are atomically distributed on the Cu surface.”

Figure R13. Characterization of 2.5% silica-loaded Cu catalyst after CO₂RR. a HAADF-STEM image. **b, c** EDX elemental mapping of Cu (**b**) and Si (**c**). **d** The Si 2p XPS spectrum. **e** EDX spectrum of the Cu-SiO_x catalyst collected from the area of (**a**).

Figure R14. Characterization of 5% silica-loaded Cu catalyst after CO₂RR. a HAADF-STEM image. **b, c** EDX elemental mapping of Cu (**b**) and Si (**c**). **d** The Si 2p XPS spectrum. **e** EDX spectrum of the Cu-SiO_x catalyst collected from the Area #1 indicated in (**a**).

3. Evidenced by the CO₂RR performances, creating the new Cu-Si sites enhanced the activity and product selectivity. The new sites render the formation of the key intermediates more easily with lower activation energy. However, the creating of the step sites would also probably just improve the accessibility of the transported CO₂. The apparent activation energy of bare Cu as well as the modified catalysts would give more direct proof.

We now more clearly show that the total current density and hydrogen partial current density of Cu-SiO_x catalysts are similar to those of the bare Cu catalysts (**Figure R15**), suggesting that CO₂ mass transport does not affect the CO₂RR on both bare Cu and the modified catalysts. Instead, the formation of Cu-SiO_x interface favours the adsorption of the OCCOH* intermediate and drives the reaction pathway toward ethylene production. The significant increase in ethylene selectivity on the modified catalyst surface enhances the ethylene partial current density, though the total current density remains unchanged with vs. without silica loading.

We now have clarified this point in our revised manuscript (**Page 10, Line 213**):

“We observed that the total and hydrogen current densities do not vary significantly in the range of silica loadings tested (**Supplementary Fig. 32**), implying similar physical availability (i.e. mass transport) of CO₂.”

Figure R15. CO₂RR performance of different Cu catalysts in a MEA electrolyzer with 0.1 M KHCO₃ anolyte. **a, b** Total current densities (**a**) and H₂ (**b**) current densities on different Cu catalysts with Si loadings in the range of 0% and 5%. Error bars are means ± SD (n = 3 replicates).

Reviewer #3

In this manuscript, the authors report an oxide doping strategy based on silica to create abundant Cu-SiO_x interface stepped sites, assisting CO₂ activation and the formation of the OCCOH intermediate. The result is interesting. This strategy provides a new possibility for enhancing the electrochemical activity and selectivity for CO₂ reduction to ethylene. However, there are still some important issues that should be well addressed before publication.*

Comment 1: The results show that a Cu-SiO_x catalyst with a Si loading of ~3% increases the efficiency of CO₂-to-ethylene conversion. The reason for this result should be discussed. I think excess SiO_x is unfavorable, which would reduce the accessibility of CO₂ to the active site. It is very important to know the coverage ratio of SiO_x on Cu.

To investigate the catalyst structure-reactivity relationship, we performed additional EXAFS analyses at the Si K-edge of various Cu-SiO_x catalysts with the silica loadings in the range of 0% and 5%. We then developed different materials models to simulate the experimental EXAFS spectra (**Figure R16**). From the Si K-edge EXAFS fitting (**Figure R17**), we found that while increasing the silica loading up to 2.5% promotes stable Cu⁰-Si²⁺ sites for enhanced ethylene formation, a further increase in the silica loading to 5% diminishes the Cu-SiO_x interface sites, reducing the ethylene selectivity from CO₂ reduction.

Using ICP-OES, we determined the silica loadings of different Cu-SiO_x-M catalysts (M = 1, 2.5 or 5 which represents the nominal silica loading percentage) to be 0.8%, 2.1% and 4.7% (**Table R2**). We then built different DFT models (**Figure R18**) with the silica loadings similar to experimental values, and we calculated the silica coverages on Cu to be ~1/16, ~2/16 and ~3/16 monolayers for the Cu-SiO_x-1, Cu-SiO_x-2.5 and Cu-SiO_x-5, respectively.

We show in **Figure R19** that the total and H₂ current densities do not change significantly with varying silica loadings ranging from 0% to 5%, indicating that the physical accessibility of CO₂ to the active site is similar at different silica loading conditions.

We have included new results and discussions in the revised manuscript:

Page 8, Line 175: “We fit the Si K-edge EXAFS spectra of different Cu-SiO_x catalysts with the relevant DFT models having the similar silica loadings. We found that the Si local structures (i.e. coordination number and bond distance) extracted from the EXAFS fitting align well with those obtained from the DFT modeling (**Fig. 3c – e, Supplementary Fig. 30 and Supplementary Table 3**), indicating general agreement between experimental and computational trends herein. From EXAFS fits, we also found a decrease in the Si-Cu coordination from 3 to 2 with the silica loading increasing from 2.5% to 5% (**Fig. 3f**), signifying that an increase in the silica loading to 5% diminishes the interaction between Si and Cu. Increasing the silica loading from 2.5% to 5% also results in a stronger Si-O-Si layer, evidenced by an increase in the Si-O coordination

from 2 to 3, an increase in the Si-Si coordination from 1 to 2, and a $\sim 7\%$ contraction in the Si-Si bond from 3.0 Å to 2.8 Å (**Fig. 3f** and **3g**).

Taken together, these results suggest that while Cu remains a metallic state in the Cu-SiO_x catalyst, the Si species preserves an oxidation state of +2 at the Cu surface. An increase in the silica loading from 1% to 2.5% promotes the stable Cu⁰-Si²⁺ interface sites, whereas a further increase in the Si loading to 5% diminishes the Si and Cu interaction and forms a strong Si-O-Si layer, destabilizing the Cu-SiO_x interface.”

Page 9, Line 202: “In a silica loading range of 0% – 5% on Cu, we found a strong correlation between the silica loading and ethylene formation (**Fig. 4b** and **Supplementary Fig. 31**). We observed a gradual increase in FE_{ethylene} and a decrease in FE_{CO} with increasing Si loading up to 2.5%, whereas we noted a decrease in FE_{ethylene} with a further increase in the Si loading from 2.5% to 5%, accompanying an enhancement in FE_{CO}. The volcano relation between FE_{ethylene} and the silica loading agrees well with the volcano correlation between the formation energy of OCCOH* and the silica loading ranging from 0% to 5% (which is equivalent to a silica coverage of 0 – 3/16 ML) predicted by DFT calculations (**Fig. 1b**).

The drop in FE_{ethylene} with the silica loading of 5% is attributable to the formation of a strong Si-O-Si layer, which diminishes the interaction between Cu and silica, creating a weak Cu-SiO_x interface to disfavor the OCCOH* adsorption toward ethylene formation (**Fig. 1** and **3**). We observed that the total and hydrogen current densities do not vary significantly in the range of silica loadings tested (**Supplementary Fig. 32**), implying similar physical availability (i.e. mass transport) of CO₂.”

Figure R16. Si K-edge EXAFS fitting of Cu-SiO_x catalysts. a-c Si K-edge EXAFS spectra (open circles) and corresponding fitting curves (solid red lines) of various Cu-SiO_x catalysts at a silica loading of 1% (a), 2.5% (b) and 5% (c). d-f Material models applied for EXAFS fitting, at a theoretical silica loading of 1.6% in (d), 3.1% in (e) and 4.7% in (f).

Figure R17. Si K-edge EXAFS fitting of Cu-SiO_x catalysts. **a, b** Simulated coordination number (**a**) and Si-Si bond distance (**b**) of different Cu-SiO_x catalysts extracted from their Si K-edge EXAFS.

Table R2. Elemental composition of Si and Cu in different Cu-SiO_x catalysts determined by inductively coupled plasma optical emission spectrometry (ICP-OES).

Sample ID	Si (mg L ⁻¹)	Cu (mg L ⁻¹)	Si:Cu ratio
Cu-SiO _x -1	1.57	444.32	0.008:1
Cu-SiO _x -2.5	3.70	398.43	0.021:1
Cu-SiO _x -5	8.01	385.75	0.047:1

Figure R18. DFT models built for the Si K-edge EXAFS fittings of Cu-SiO_x-1 (1/16 ML), Cu-SiO_x-2.5 (2/16 ML) and Cu-SiO_x-5 (3/16 ML).

Figure R19. CO₂RR performance of different Cu catalysts in a MEA electrolyzer with 0.1 M KHCO₃ anolyte. a, b Total current densities (a) and H₂ (b) current densities on the bare Cu and various Cu-SiO_x catalysts. Error bars are means ± SD (n = 3 replicates).

Comment 2: The authors declare a uniform distribution of Si in Cu, as evidenced by elemental mapping. However, the morphology of the Cu-SiO_x catalyst is unclear, especially the assembly style of Cu and SiO_x. The size and shape of SiO_x may be key to the performance of the catalyst. The catalyst should be characterized by more techniques.

We have further characterized the Cu-SiO_x catalyst using high-resolution HAADF-STEM (**Figure R20 - R22**). Results show that silica species are homogeneously distributed on the Cu surface at the atomic scale with no visible aggregation of silica, consistent with the EXAFS fitting results indicated above in our reply to the comment 1 of the Reviewer.

We now have clarified these structural details in our revised manuscript (**Page 7, Line 150**):

“Transmission electron microscopy and scanning electron microscopy images show nanoparticles in an aggregated form and a mean size of ~50 nm. The Cu morphology is unchanged with silica loading (**Fig. 2a** and **Supplementary Fig. 20**). The catalysts with/without silica loading lead to similar Cu patterns, seen in X-ray powder diffraction (XRD) measurements (**Supplementary Fig. 21**). We examined and compared the structure of the Cu-SiO_x catalyst before and after CO₂RR using high-angle annular dark-field scanning transmission electron microscopy (HAADF-STEM) and energy-dispersive X-ray (EDX) mapping. We found that the Si species are evenly distributed on the Cu catalyst surface, and that the structure of Si species remains intact before and after the reaction, showing an oxidation state of +2 as revealed by X-ray photoelectron spectroscopy (XPS, **Fig. 2b – f** and **Supplementary Fig. 22 – 25**). Using HAADF-STEM and XPS, we also characterized and compared Cu-SiO_x catalysts with the silica loadings of 2.5% (**Fig. 2**) and 5% (**Supplementary Fig. 26**), and we found similar silica structures in both samples and detected no aggregation of silica, suggesting that silica clusters are

atomically distributed on the Cu surface.

To investigate the electronic interaction between silica and Cu, we performed X-ray absorption spectroscopy (XAS) measurements at the Cu K-edge and Si K-edge. In-situ X-ray absorption near-edge structure (XANES) at the Cu K-edge reveals the metallic nature of Cu active sites (**Fig. 3a**), corroborated by the extended X-ray absorption fine structure (EXAFS, **Supplementary Fig. 27**). XANES at the Si K-edge reveals an average +2 oxidation state of the Si species based on the dipole-allowed ($1s \rightarrow 2p$ transition) absorption onset energy position (**Fig. 3b**). Increasing the silica loading on Cu from 1% to 5% results in an enhancement of the Si K-edge whiteness intensity and a blue shift of the absorption onset (**Supplementary Fig. 28**), indicating that the Si oxidation state increases with increasing silica loading, in line with our DFT predictions (**Fig. 1a**).

The Si K-edge EXAFS shows a strong peak at 1.3 Å, attributable to the Si-O bond in the first shell of Si atoms (**Supplementary Fig. 29**). We fit the Si K-edge EXAFS spectra of different Cu-SiO_x catalysts with the relevant DFT models having the similar silica loadings. We found that the Si local structures (i.e. coordination number and bond distance) extracted from the EXAFS fitting align well with those obtained from the DFT modeling (**Fig. 3c – e, Supplementary Fig. 30 and Supplementary Table 3**), indicating general agreement between experimental and computational trends herein. From EXAFS fits, we also found a decrease in the Si-Cu coordination from 3 to 2 with the silica loading increasing from 2.5% to 5% (**Fig. 3f**), indicating that an increase in the silica loading to 5% diminishes the interaction between Si and Cu. Increasing the silica loading from 2.5% to 5% also results in a stronger Si-O-Si layer, evidenced by an increase in the Si-O coordination from 2 to 3, an increase in the Si-Si coordination from 1 to 2, and a ~7% contraction in the Si-Si bond from 3.0 Å to 2.8 Å (**Fig. 3f and 3g**).

Taken together, these results suggest that while Cu remains in a metallic state in the Cu-SiO_x catalyst, the Si species preserve an oxidation state of +2 on the Cu surface. An increase in the silica loading from 1% to 2.5% promotes the stable Cu⁰-Si²⁺ interface sites, whereas a further increase in the Si loading to 5% diminishes the Si and Cu interaction and forms a strong Si-O-Si layer, destabilizing the Cu-SiO_x interface.”

Figure R20. Characterization of 2.5% silica-loaded Cu catalyst after CO₂RR. a, b TEM image (a) and HAADF-STEM image (b). c, d EDX elemental mapping of Cu (c) and Si (d). e EDX spectrum of the Cu-SiO_x catalyst collected from the area of (b). f, g XPS spectra of Cu 2p (f) and Si 2p (g). Results show that Si species are homogeneous distributed on the Cu catalyst and maintain an oxidation state of +2 after CO₂RR.

Figure R21. Characterization of 2.5% silica-loaded Cu catalyst after CO₂RR. a HAADF-STEM image. b, c EDX spectra of the Cu-SiO_x catalyst collected from the surface Area #1 and bulk Area #2 indicated in (a), suggesting that Si species are distributed on the Cu surface rather than in the Cu bulk.

Figure R22. Characterization of 2.5% silica-loaded Cu catalyst after CO₂RR. a, b HAADF-STEM images at different length scales. **c, d** High-resolution electron energy loss spectra of Cu (**c**) and Si (**d**) collected from the area of (**b**), showing that no significant aggregation of Si species is observed.

Comment 3: It is better to give the in-situ characterization for the local atomistic structure of the material during electrocatalysis.

We confirm that all Cu K-edge (~8979 eV) XAS spectra in this work were performed *in situ*, at the 9BM beamline of the Advanced Photon Source (9BM@APS, <https://www.aps.anl.gov/Spectroscopy/Beamlines/9-BM>). We have now clarified this aspect in the revised manuscript. Details of the in-situ XAS cell design and testing were included in our previous work (Ref 37: *Nat. Commun.* **9**, 4614 (2018)).

Since the absorption energy of Si K-edge (~1839 eV) is below the energy range of 9BM beamline (2.1 – 24 keV) and the soft X-ray absorption measurement requires ultrahigh vacuum system, we conducted the Si K-edge experiments at the Soft X-ray Microcharacterization Beamline of the Canadian Light Source (SXRMB@CLS, 1.7 – 10 keV, https://www.lightsource.ca/beamlines/details/soft_xray_microcharacterization_beamline_sxrmb.html). However, the sample testing with a mail-in service was the only option at CLS due to a partial lockdown at the moment. Due to this circumstance, we instead performed ex-situ Si K-edge testing. Samples after CO₂RR were rinsed with DI water, dried using N₂ gas and then stored under vacuum at room temperature overnight, and sealed with Kapton tape in N₂-filled glove box. Ex-situ Si K-edge measurements show that the Si species in different Cu-SiO_x catalysts preserve a ~+2 oxidation state after CO₂RR and no oxidation of Cu is observed from synchronous ex-situ Cu K-edge XAS testing at SXRMB beamline (**Figure R23**), suggesting the nature of Cu-SiO_x interface sites for CO₂RR.

We have added the following information in revised manuscript for clarity:

Page 13, Line 303: “In-situ XAS measurements at the Cu K-edge were carried out at the 9BM beamline of Advanced Photon Source (APS, Argonne National Laboratory, Lemont, Illinois, USA) using a home-made flow cell³⁷. Ex-situ Si K-edge experiments were conducted at the Soft X-ray Microcharacterization Beamline (SXRMB) of the Canadian Light Source (CLS, Saskatoon, Saskatchewan, Canada) by sealing samples in Kapton tape after the CO₂RR testing and vacuum drying. Ex-situ XAS measurements at the Cu K-edge were also recorded at the SXRMB beamline together with ex-situ Si K-edge testing to exclude material oxidation from ex-situ measurements (**Supplementary Fig. 36**). Fluorescence yield was recorded using silicon drift detectors at both the 9BM and SXRMB beamlines. XANES data were analyzed using Athena software included in a standardized IFEFFIT package³⁸. EXAFS fittings were performed using the FEFF software included in the IFEFFIT package³⁹.”

Page 23, Line 516: “In-situ Cu K-edge XAS was performed during CO₂RR whereas ex-situ Si K-edge XAS was measured upon completion of CO₂RR at a full cell potential of -4.1 V in 0.1 M KHCO₃.”

Figure R23. Ex-situ XAS characterization of the Cu-SiO_x catalyst at a silica loading of 2.5%. a Si K-edge XANES of the Cu-SiO_x catalyst, Si wafer and SiO₂. **b, c** Cu K-edge XANES (b) and EXAFS (c) of the Cu-SiO_x catalyst, Cu foil and Cu₂O.

Comment 4: The results in Fig. 4a-c should be discussed in more details. An explanation for the FE changes of different products is needed.

We have added more discussion in the revised manuscript (**Page 10, Line 231**):

“To challenge the versatility of the Cu-SiO_x catalyst, we assessed CO₂RR performance using dilute input gas streams with CO₂ concentrations in a range typical of flue gas³². We tested the CO₂RR performance of the Cu-SiO_x-2.5 catalyst across a cell potential window of -3.4 V ~ -4.2 V by co-feeding CO₂ and N₂ gas feeds with the CO₂

concentrations ranging from 10% to 100%. We found that reducing the CO₂ concentration induces a slight decrease in current density (**Supplementary Fig. 34**), due to reduced CO₂ availability at the catalyst layer, evidenced by the increasing H₂ current densities at reduced CO₂ concentrations.

Regulating the CO₂ concentration in the full-cell potential window of -3.6 V ~ -4.2 V results in a high FE_{ethylene} of >50% with varied FE_{CO} and FE_{H2} (**Fig. 5a – c**). At -3.6 V, we achieved 52% FE_{ethylene} (15% FE_{CO}) with 10% CO₂ feed, in contrast to 37% FE_{ethylene} with an increase of FE_{CO} to 26% using 100% CO₂ at similar conditions. The sharp increase of FE_{CO} at increasing CO₂ concentrations can be attributed to excessive local CO₂ in the catalyst layer, blocking active catalyst surface sites for CO dimerization toward ethylene formation and promoting instead CO production¹⁷.

Decreasing the potential to -4.2 V leads to a high FE_{ethylene} of 64% (13% FE_{H2}) with 100% CO₂ feed whereas the FE_{ethylene} decreases to 30% with a dramatic increase of FE_{H2} to 58% using 10% CO₂ at the same applied potential. A shift of product selectivity to H₂ at low CO₂ concentrations and at high negative potentials can be attributed to the CO₂ mass transport limitations (**Fig. 5c** and **Supplementary Fig. 34**). Take together, these findings show that FE_{ethylene} increases when we constrain CO₂ availability, which can be linked to modulation of the surface coverage of *CO₂, *CO, and *H at the catalyst layer¹⁷. As a result, we achieve high FE_{ethylene} from CO₂RR by tuning the local CO₂ concentration and applied potential/current density. Tuning conditions for high FE_{ethylene} and low cell potential, we achieve an 18% full-cell energy efficiency (EE) toward ethylene production across a broad range of CO₂ concentrations, 10 – 100% (**Fig. 5d**).”

We thank the three reviewers for thoughtful input that motivated deep characterization studies that support the key findings of the manuscript.

Reviewer #1 (Remarks to the Author):

With respect to the previous version, the current manuscript includes the structural characterization of the electrocatalysts, in some cases also performed in situ, with the goal to provide mechanistic insights that complement the theoretical study.

As summarized in the abstract, "by means of theory calculations and X-ray absorption spectroscopy they find that anchoring silica on Cu provides abundant CuO-Si₂⁺ step sites".

However, the main conclusion of this contribution, that is the existence of CuO-Si₂⁺ sites is still not explored/verified experimentally in this version, since only the Cu K edge XANES was measured in situ, whereas the Si K edge XANES was done ex-situ.

Particularly for electro-catalysis, the importance of measuring the electronic structure in situ was shown in many contributions. By allowing the electrocatalysts to reach the open circuit potential, before disassembling the cell, a surface speciation will form completely different from that existing under conditions of CO₂ reduction. This referee thinks that these data should be certainly provided otherwise their claim is weak and probably also incorrect as at least another mechanism could be involved.

This referee is not positive about this contribution. One should ask: What is the novelty here?

i) Is it the strategy to enhance the selectivity by including chemisorption sites for CO₂, which remain stable at low potential (IUPAC definition) and increases the chances of C-C coupling? Well this is not really new and the authors themselves have published many contributions in which they have attempted to do that. In this contribution they are using a less reducible element but this work appears to me incremental and not very novel. The authors are neglecting earlier literature on the use of Si/C in CO₂ ER.

ii) Is it the fact that the authors are discussing now a gas-phase approach for the CO₂ ER? Again, there is a huge focus on the developments made in this group (heavy use of self-citation or comparison with own work table 4), while no mention of the very early literature in which this approach was proposed for the first time. And while this referee appreciates that there is a limited number of citations that can be included, it still makes me wonder if the authors are aware of the broader context of their research and when looking at this it make me think of the limited novelty of this work. The authors state that their results outperform previous work in their group within the same approach but this referee is not convinced that this statement can be generalized (see for example early work of Sammells (FE of 53% for ethylene at a density of 667 mA cm⁻²)) without including the comparison of the energy efficiency of the process. Why is this element completely neglected? The readership will not even be able to compare their results with this contribution since the electro-catalytic data are reported vs the overall cell potential and not vs a reference electrode (for specialist readership).

The understanding of the role of Si is interesting, however should be carried out more extensively and probably, is material for a full paper rather than a communication.

There are also a couple of inaccuracies in definitions and inconsistencies and this referee would recommend that the authors reflect upon the following points to deliver a scientifically valuable contribution.

i) The term doping implies the addition of normally <1000 ppm of a heteroatom, which has an electronic effect on the Cu. Here the authors use a larger amount of Si. They also refer to the effect of Si₂⁺ as favouring step site. Such a large amount of Si₂⁺ and such an effect is generally referred in

catalysis as structural promoters.

ii) Doping implies an electronic modification, which the authors are not able to clearly see in the in situ Cu K edge, being it predominantly metallic. I would have attempted a fitting of the Cu K edge XANES in Fig 3a and EXAFS in Fig 27 of the supplementary information to identify contribution from Cu⁺ to prove an electronic effect?

iii) The identification of Si²⁺ is questionable: based on Si2p XPS binding energy a BE at 102-103 eV (Fig. 2 of the main text and Figure 22 of the supporting information) one would conclude an higher oxidation state as for example Si³⁺ species or thin layer Si⁴⁺ on metal; the authors also state Page 8 line 169-170) : XANES at the Si K-edge reveals an average +2 oxidation state of the Si species based on the dipole-allowed (1s → 2p transition) absorption onset energy position (Fig. 3b). However extracting the oxidation state of the metal centre from a transition strongly depending on ligands effect is dangerous. It is also not clear how the conclusion of the oxidation state was drawn. The authors are encouraged to look into the relevant literature and refer critically to it when they attempt to analyse their results.

iv) At page 8 Line 171-173 The authors state: "Increasing the silica loading on Cu from 1% to 5% results in an enhancement of the Si K-edge white-line intensity and a blue shift of the absorption onset (Supplementary Fig. 28), indicating that the Si oxidation state increases with increasing silica loading, in line with our DFT predictions (Fig. 1a)." . an enhancement of the white line means an increase of the density of unoccupied state which might have predominantly ligand character, therefore this statement can be misleading and scientifically incorrect. A more critical analysis necessary for a Nature paper.

v) The authors have attempted to identify the geometric configuration of the Si sites and the location.

a. At page 8 Line 162-163, the authors state: suggesting that silica clusters are atomically distributed on the Cu surface. A cluster cannot be atomically distributed, as it is a cluster.

b. The HAADF-STEM and EDX analysis is also performed in an ex situ fashion and therefore conclusions from these data should be drawn but with a critical consideration on the impact of the potential removal. A part of this, the analysis shows that the Si is homogeneously distributed before and after reaction.

i. TEM is local, is there a statistical analysis?

ii. In Figure 23 of the supplementary information it is indicated that the EDX was performed after the reaction in the area of Figure 2b, does it mean that an identical location study was performed? If this is the case, then the technical details of this experiment should be described.

iii. Supplementary Fig. 24 suggests that Si species are distributed on the Cu surface rather than in the Cu bulk., so some Si segregation occurs. In Figure 25 of the SI an homogeneous distribution of Si is reported, however it is not clear from which part of the particle in Fig 25a this is taken (bulk Si-free or Si-containing surface?). These data should be presented better. Also, in elemental mapping in Fig 25 c and d the pixel size seems much smaller than the actual resolution in Figure 25b. How small the step size can be in the mapping to integrate the signal? It looks to me more like a mapping of the noises rather than the signal. One would expect that the Cu signal would increase where the atoms are in Figure 25b. However atomically resolved EDX mapping is very difficult to achieve. But this shows also the technical flaw of this measurement. Are these measurements needed after all? I am getting back to the point of atomically dispersed clusters mentioned by the authors, what does actually mean? Is it this all about excluding Si-Si direct bonding?

iv. Page 8 Line 183-191. The authors state that the increase in the silica loading to 5% diminishes the interaction between Si and Cu. Increasing and forms a strong Si-O-Si layer, destabilizing the Cu-SiOx interface. This referee think that this should not be explained as diminishing of the interaction between Si-Cu, but a change from a 2D-thin layer of Si oxide into a 3-D overlayer.

Overall, this referee thinks that the paper is still not in a shape to be published in this journal. The nature of Si under in situ conditions and the effect of it on the Cu has to be clarified (electronic and geometric). If the Si role is to enhance the step sites this should be counted. Also the theoretical study is based on a mechanism of reaction proposed earlier and considering energetics of the intermediates. This referee thinks that the role of Si can be very different from what the authors are suggesting but other techniques more sensitive to light elements should be applied to uncover this.

Reviewer #2 (Remarks to the Author):

I find the authors made much efforts in revision and supplemented more evidence for the raised questions. I believe the characterizations and the calculations represent the strong part. Additionally, I noticed in the experimental section, that the AEM was activated for at least 24 hours in 1 M KOH and rinsed with deionized (DI) water before test. Considering the etching feature of KOH to silica, I would ask whether rinsing is sufficient to remove the free KOH confined inside. Also I suggest the authors to discuss whether the enriching strategy of the interface sites by silica doping is a general way.

Reviewer #3 (Remarks to the Author):

The problems have been well solved. I suggest its publication.

Reviewer 1

With respect to the previous version, the current manuscript includes the structural characterization of the electrocatalysts, in some cases also performed in situ, with the goal to provide mechanistic insights that complement the theoretical study.

As summarized in the abstract, “by means of theory calculations and X-ray absorption spectroscopy they find that anchoring silica on Cu provides abundant Cu⁰-Si²⁺ step sites”. However, the main conclusion of this contribution, that is the existence of Cu⁰-Si²⁺ sites is still not explored/verified experimentally in this version, since only the Cu K edge XANES was measured in situ, whereas the Si K edge XANES was done ex-situ.

Particularly for electro-catalysis, the importance of measuring the electronic structure in situ was shown in many contributions. By allowing the electrocatalysts to reach the open circuit potential, before disassembling the cell, a surface speciation will form completely different from that existing under conditions of CO₂ reduction. This referee thinks that these data should be certainly provided otherwise their claim is weak and probably also incorrect as at least another mechanism could be involved.

Due to the lockdown, we were unable to perform additional in-situ XAS measurements, and we sought an alternative approach to address the spirit of the reviewer’s concern.

To assess speciation *during* CO₂ reduction, we conducted in-situ Raman spectroscopy measurements on different Cu catalysts at a potential range of -1.5 V – -2.4 V (**Figure R1-2**, and **Table R1** – further increasing the cell potential resulted in mass production of gas bubbles at the catalyst surface which blocked Raman signal).

A comparison between the bare Cu and the Cu-SiO_x-2.5 catalysts shows that a silica addition promotes the Cu-O-Si interaction (**Figure R1**), enhancing CO coverage at the Cu surface evidenced by an additional Cu-CO peak at 365 cm⁻¹ (starting at -1.8 V, ref. 33: Surf. Sci. 287-288, 104-109 (1993)). The Cu-O-Si interaction peak remains at all silica loadings applied. However, the onset potentials of Raman peak at 365 cm⁻¹ shift to -2.0 V and -2.2 V for Cu-SiO_x-1 and Cu-SiO_x-5, respectively, suggesting that the CO content on the Cu-SiO_x-1 and Cu-SiO_x-5 catalysts is lower than that on the Cu-SiO_x-2.5 catalyst.

These new in-situ Raman findings indicate that Cu-SiO_x interface sites function as active sites. The increase of silica loading from 0% to 2.5% facilitates CO₂ activation and enhances the CO coverage on Cu, whereas a further increase of silica loading to 5% decreases the CO content on Cu. This effect of silica loading on CO coverage on the Cu surface is consistent with our DFT predictions of the formation energy of OCOH* – an intermediate from CO₂-to-CO conversion – at the Cu-SiO_x interface with various silica loadings. The Cu-SiO_x catalyst with

optimal CO coverage at a silica loading of 2.5% accelerates the C-C coupling by lowering the formation energy of OCCOH* at the Cu-SiO_x interface, facilitating the formation of ethylene (Ref. 34: Nat. Catal. 4, 20-27 (2021)).

We have included these new results and discussions in the revised manuscript:

- **Page 10, Line 211:** “To understand the role of silica in promoting ethylene production at the Cu-SiO_x catalyst, we performed in-situ Raman spectroscopy (**Supplementary Table 4**). At a range of applied cell potentials from -1.5 V to -2.0 V (**Fig. 4a – b**), we observed one band at 280 cm⁻¹ on the bare Cu electrode starting from -1.8 V, attributable to the Cu-CO frustrated rotation at metallic Cu surface^{32,33}. On the Cu-SiO_x-2.5 electrode, the same band emerged at the cell potential of -1.8 V, together with the emergence of a new band at 365 cm⁻¹. This additional band is due to the Cu-O stretch at metallic Cu surface, indicating that a silica addition increases CO coverage at the Cu surface³³. In addition, we also observed a strong band at 530 cm⁻¹ on the Cu-SiO_x-2.5 electrode throughout the cell potentials applied, which we attribute to the presence of Cu⁺ promoted by the Cu-O-Si interaction³².

To further assess the role of silica, we carried out in-situ Raman measurements on the Cu-SiO_x-1 and Cu-SiO_x-5 electrodes (**Supplementary Fig. 35**). While we observed the stable Cu-O-Si band at 530 cm⁻¹ in both cases throughout the cell potentials applied, the Cu-O stretch band at 365 cm⁻¹ emerged at a cell potential of -2.0 V on Cu-SiO_x-1 and -2.2 V on Cu-SiO_x-5. This shift in onset potential of the 365 cm⁻¹ band indicates the tuning of CO content on Cu in response to silica loading. The Cu-SiO_x-2.5 catalyst shows the lowest onset potential of -1.8 V, suggesting the highest CO concentration at the Cu-SiO_x-2.5 catalyst surface during CO₂RR. This finding is consistent with our DFT predictions of the formation energy of OCOH* – an intermediate in CO₂-to-CO conversion – at the Cu-SiO_x interface (**Supplementary Fig. 5 – 7**). The Cu-SiO_x catalyst with optimal CO coverage at a silica loading of 2.5% accelerates the C-C coupling by lowering the formation energy of OCCOH* at the Cu-SiO_x interface, facilitating the formation of ethylene (**Fig. 1b**)³⁴.”

- **Page 17, Line 384:** “In-situ Raman experiments were performed using a Renishaw inVia Raman microscope in a home-made flow cell (**Supplementary Fig. 44**) with a water immersion objective. During testing, CO₂ with a constant flow rate of 20 s.c.c.m. flowed through the gas compartment, whereas 0.1 M KHCO₃ solution with a flow rate of 0.2 mL min⁻¹ flowed through the electrolyte compartment. A Pt wire was used as the counter electrode, and a 785-nm laser was used for Raman testing. Signals were recorded using a 5-s integration and averaging ten scans.”

Figure R1. In-situ Raman characterizations. a,b In-situ Raman spectra of Cu (a) and Cu-SiO_x-2.5 (b) under different cell potentials.

Figure R2. In-situ Raman characterizations. a,b In-situ Raman spectra of Cu-SiO_x-1 (a) and Cu-SiO_x-5 (b) under different cell potentials.

Table R1. Current densities recorded at in-situ Raman measurements for different Cu catalysts under various applied cell potentials.

Cell Potential (V)	Cu (mA cm ⁻²)	Cu-SiO _x -2.5 (mA cm ⁻²)	Cu-SiO _x -1 (mA cm ⁻²)	Cu-SiO _x -5 (mA cm ⁻²)
-1.5	4.7	5.2	N/A	N/A
-1.6	6.2	6.8	7.6	7.0
-1.7	8.0	8.9	10.0	9.1
-1.8	10.0	11.3	12.6	11.5
-1.9	12.2	14.1	15.6	14.2
-2.0	14.7	17.1	19.2	17.4

-2.2	N/A	N/A	26.2	23.6
-2.4	N/A	N/A	33.1	30.6

This referee is not positive about this contribution. One should ask: What is the novelty here?

i) Is it the strategy to enhance the selectivity by including chemisorption sites for CO₂, which remain stable at low potential (IUPAC definition) and increases the chances of C-C coupling? Well this is not really new and the authors themselves have published many contributions in which they have attempted to do that. In this contribution they are using a less reducible element but this work appears to me incremental and not very novel. The authors are neglecting earlier literature on the use of Si/C in CO₂ ER.

ii) Is it the fact that the authors are discussing now a gas-phase approach for the CO₂ ER? Again, there is a huge focus on the developments made in this group (heavy use of self-citation or comparison with own work table 4), while no mention of the very early literature in which this approach was proposed for the first time. And while this referee appreciates that there is a limited number of citations that can be included, it still makes me wonder if the authors are aware of the broader context of their research and when looking at this it make me think of the limited novelty of this work. The authors state that their results outperform previous work in their group within the same approach but this referee is not convinced that this statement can be generalized (see for example early work of Sammells (FE of 53% for ethylene at a density of 667 mA cm⁻²)) without including the comparison of the energy efficiency of the process. Why is this element completely neglected? The readership will not even be able to compare their results with this contribution since the electro-catalytic data are reported vs the overall cell potential and not vs a reference electrode (for specialist readership).

We have not found in the literature any experimental evidence of Si or Si/C in promoting CO₂ electroreduction. However, Si-based materials have been applied to facilitate CO₂ hydrogenation via thermo-photocatalysis (Ref. 26: Nat. Catal. 2, 46-54 (2019); ref. 27: Nat. Commun. 7, 12553 (2016)). Those reports suggest that when reacting with CO₂, Si deactivates quickly by forming surface silanols (Si-OH) and siloxane (Si-O-Si) groups, indicating that Si itself on Cu is not stable in CO₂ hydrogenation and Cu-SiO_x instead is a more stable catalyst structure in promoting CO₂ electroreduction.

The novelty of this work comes from the use of silica in CO₂ electroreduction (materials innovation) and the application of that approach in an electrolyzer (systems advancement). The membrane electrode assembly (MEA) electrolyzer can offer high stability and energy efficiency at practical production rates – all of which are essential for the practical application of CO₂RR. The advantages of MEA operated in neutral media over alkaline flow electrolyzer used by Sammells et al. (Ref. 6: J. Electrochem. Soc. 137, 607-608 (1990)) have been directly

compared and reported not only by our group (e.g. ref. 22: *Nature* 577, 509-513 (2020); ref. 10: *Joule* 3, 2777-2791 (2019)) but also from other research groups in the field (e.g. ref. 12: *Science* 365, 367-369 (2019); ref. 11: *Joule* 3, 240-256 (2019)). The use of an MEA electrolyzer to optimize CO₂RR performance has also been reported recently (e.g. ref. 12: *Science* 365, 367-369 (2019); ref. 13: *Joule* 3, 1487-1497 (2019); ref. 14: *Nat. Energy* 6, 46-53 (2021)).

Our applied goal in this work was to achieve selective and active CO₂ reduction to ethylene in an MEA electrolyzer, with a potential to operate at cheap and dilute CO₂ concentrations. While our recent effort has achieved a 64% ethylene selectivity from CO₂RR in MEA (Ref. 22: *Nature* 577, 509-513 (2020)), the copper-molecule strategy is limited by insufficient active sites, leading to an ethylene productivity of 80 mA cm⁻².

In this work, we introduce the silica cluster as a structural promoter to construct active, selective and stable Cu-SiO_x interface sites, with the aim to accelerate the C-C coupling reaction to produce ethylene from CO₂ reduction with high productivities. While abundant CO₂ mass transfer (enabled by MEA) at the bare Cu surface blocks active sites for C-C coupling to produce ethylene (Ref. 17: *Joule* 4, 1104-1120 (2020)), we show here (with DFT and experimental findings) that a silica loading of ~3% on Cu assists CO₂ activation and increases ethylene production. A further increase of silica loading to 5% diminishes Cu-SiO_x interaction and blocks Cu active sites, resulting in mass production of CO instead of ethylene. The unique role of silica is due to the acceptance and back-donation of electrons (i.e. the formation of Si-C and Si-O bonds shown in Fig. 1c) to enable efficient CO₂ activation and C-C coupling at the Cu-SiO_x interface.

As a result, we achieve a 65% ethylene selectivity at an ethylene productivity of 215 mA cm⁻² using a 2.5% silica-loaded Cu catalyst. The Cu-SiO_x-2.5 catalyst based MEA electrolyzer maintains high ethylene selectivities of >60% at high reaction rates (>120 mA cm⁻²) even under dilute CO₂ feedstocks (10% concentration); and features sustained operation over 50 hours.

We have included additional discussions in our revised manuscript to highlight the novelty of this work:

- **Page 3, Line 47:** “Gas diffusion electrodes (GDEs) embedded in alkaline flow cell electrolyzers have enabled selective CO₂-to-ethylene conversion at industrial-level production rates^{6,7}. However, CO₂ utilization remains low due to the rapid reaction of CO₂ molecules and hydroxide ions in these systems². In addition, direct contact between GDE and aqueous electrolyte leads to electrode flooding and catalyst deactivation^{8,9}.”

The membrane electrode assembly (MEA) electrolyzer, with a direct cathode:membrane:anode contact, offers a platform that is more stable than alkaline flow cell electrolyzers¹⁰⁻¹². A low cell resistance of zero-gap MEA electrolyzers also enables the use of bicarbonate or carbonate electrolyte without sacrificing the CO₂RR efficiency^{13,14}. ”

The understanding of the role of Si is interesting, however should be carried out more extensively and probably, is material for a full paper rather than a communication.

We have revised manuscript to be a full paper, providing greater detail on the role of silica in promoting ethylene at the Cu-SiO_x interface, as requested, with details in the “Investigating the Cu-SiO_x interaction” and “Discussion” sections.

There are also a couple of inaccuracies in definitions and inconsistencies and this referee would recommend that the authors reflect upon the following points to deliver a scientifically valuable contribution.

i) The term doping implies the addition of normally <1000 ppm of a heteroatom, which has an electronic effect on the Cu. Here the authors use a larger amount of Si. They also refer to the effect of Si²⁺ as favouring step site. Such a large amount of Si²⁺ and such an effect is generally referred in catalysis as structural promoters.

We have removed the term doping, and refer instead to modulation and structural promoters.

ii) Doping implies an electronic modification, which the authors are not able to clearly see in the in situ Cu K edge, being it predominantly metallic. I would have attempted a fitting of the Cu K edge XANES in Fig 3a and EXAFS in Fig 27 of the supplementary information to identify contribution from Cu⁺ to prove an electronic effect?

With further analysis as suggested we found no trace of Cu⁺ in the Cu K-edge XANES and EXAFS results, which is consistent with the absorption onset at 8979 eV for all Cu-SiO_x samples and Cu foil. The Cu⁺ shows a distinct absorption onset at ~8980 eV (**Figure R3a**).

We instead performed in-situ Raman analysis at all Cu catalysts (**Figure R1-2**), and found the presence of Cu⁺ enabled by the silica addition, in contrast to the metallic Cu feature of bare Cu catalyst. Details are included above in our response to the first comment of the Reviewer.

The absence of Cu⁺ from the Cu K-edge XAS analysis is due to the bulk-sensitive detection of fluorescent X-rays (**Figure R3b**), in agreement with previous reports (e.g. ref. 35: ACS Catal. 9, 5035-5046 (2019)). The presence of strong Cu-O-Si interaction revealed by in-situ Raman indicates that silica locates only at the Cu NP surface, consistent with our DFT calculations and TEM results.

We have included these new results and discussions in the revised manuscript:

- Page 11, Line 234:** “In-situ X-ray absorption near-edge structure (XANES) at the Cu K-edge shows the metallic nature of Cu active sites, corroborated by the extended X-ray absorption fine structure (EXAFS, **Supplementary Fig. 36a – c**). Although the presence of Cu⁺ from the Cu-O-Si interaction is evidenced by in-situ Raman analysis, the Cu K-edge XAS recorded in fluorescence yield probes further into the bulk (**Supplementary Fig. 36d**)³⁵. The predominantly metallic Cu states of the Cu-SiO_x catalysts revealed by XAS further supports our view that silica clusters are deposited at the surface of Cu nanoparticles. This finding is also in line with the HAADF-STEM results (**Supplementary Fig. 27**).”

Figure R3. Cu K-edge XAS. **a** The first derivative spectra of different Cu-SiO_x catalysts and the Cu foil and Cu₂O standards. **b** Calculated X-ray attenuation length vs. photon energy in metallic Cu (density = 8.96 g cm⁻³). The incident angle is 45 degree (http://henke.lbl.gov/optical_constants/atten2.html).

iii) The identification of Si²⁺ is questionable: based on Si2p XPS binding energy a BE at 102-103 eV (Fig. 2 of the main text and Figure 22 of the supporting information) one would conclude an higher oxidation state as for example Si³⁺ species or thin layer Si⁴⁺ on metal; the authors also state Page 8 line 169-170) : XANES at the Si K-edge reveals an average +2 oxidation state of the Si species based on the dipole-allowed (1s → 2p transition) absorption onset energy position (Fig. 3b). However extracting the oxidation state of the metal centre from a transition strongly depending on ligands effect is dangerous. It is also not clear how the conclusion of the oxidation state was drawn. The authors are encouraged to look into the relevant literature and refer critically to it when they attempt to analyse their results.

The assignment of Si 2p_{3/2} XPS peak at 102.5 eV to Si²⁺ in this work is consistent with previous reports (e.g. ref. 31: J. Chem. Phys. 68, 1776-1784 (1978)).

The Si K-edge spectral features and absorption onset of the Cu-SiO_x catalysts echo those of molecular SiO deposited on highly oriented pyrolytic graphite (Ref. 36: Surf. Sci. 612, 77-81 (2013)), confirming the +2 oxidation state of Si species in our catalysts.

We have revised the relevant statements with this further literature support:

- **Page 11, Line 243:** “The Si K-edge XANES of the Cu-SiO_x catalysts reveals an average +2 oxidation state of the Si species (**Fig. 4c**), with absorption onset energy (1s → 2p transition) and spectral characteristics analogous to the Si K-edge XANES of the molecular SiO deposited on highly oriented pyrolytic graphite, reported elsewhere³⁶.”

iv) At page 8 Line 171-173 The authors state: “Increasing the silica loading on Cu from 1% to 5% results in an enhancement of the Si K-edge white-line intensity and a blue shift of the absorption onset (Supplementary Fig. 28), indicating that the Si oxidation state increases with increasing silica loading, in line with our DFT predictions (Fig. 1a).” . an enhancement of the white line means an increase of the density of unoccupied state which might have predominantly ligand character, therefore this statement can be misleading and scientifically incorrect. A more critical analysis necessary for a Nature paper.

We have revised the statements in the revised manuscript:

- **Page 11, Line 246:** “Increasing the silica loading on Cu from 1% to 5% results in an enhancement of the Si K-edge white-line intensity and a blue shift of the absorption onset (**Supplementary Fig. 37**). A broadening of white-line indicates an overall Si p-charge depletion due to charge redistribution at the Cu-SiO_x interface with increasing silica loadings, and a positive energy shift of Si 1s absorption onset suggests a potential increase of the Si oxidation state from the charge redistribution process.”

v) The authors have attempted to identify the geometric configuration of the Si sites and the location.

a. At page 8 Line 162-163, the authors state: suggesting that silica clusters are atomically distributed on the Cu surface. A cluster cannot be atomically distributed, as it is a cluster.

In the revised manuscript, we have now replaced “atomically” to “uniformly”.

b. The HAADF-STEM and EDX analysis is also performed in an ex situ fashion and therefore conclusions from these data should be drawn but with a critical consideration on the impact of the potential removal. A part of this, the analysis shows that the Si is homogeneously distributed before and after reaction.

We have acknowledged the limitation of ex-situ analysis of the catalyst structure in the revised manuscript:

- **Page 8, Line 170:** “We note that characterizations of catalyst morphology were performed under ex-situ conditions. Although the extent of any catalyst reconstruction due to the potential removal after CO₂RR is not expected to be major, the development of in-situ TEM capabilities will be useful to assess the impact of potential removal on catalyst reconstruction, if any.”

i. TEM is local, is there a statistical analysis?

For statistical analysis, we now include additional representative EDX mapping results at three different regions of interest of the Cu-SiO_x-2.5 (**Figure R4-6**) and Cu-SiO_x-5 (**Figure R7-9**), respectively, after CO₂RR. We find the earlier TEM results to be broadly representative of the sample.

Figure R4. HAADF-STEM image and EDX elemental mapping of 2.5% silica-loaded Cu catalyst after CO₂RR at the region of interest one. Cu is in red and Si is in green.

Figure R5. HAADF-STEM image and EDX elemental mapping of 2.5% silica-loaded Cu catalyst after CO₂RR at the region of interest two. Cu is in red and Si is in green.

Figure R6. HAADF-STEM image and EDX elemental mapping of 2.5% silica-loaded Cu catalyst after CO₂RR at the region of interest three. Cu is in red and Si is in green.

Figure R7. HAADF-STEM image and EDX elemental mapping of 5% silica-loaded Cu catalyst after CO₂RR at the region of interest one. Cu is in red and Si is in green.

Figure R8. HAADF-STEM image and EDX elemental mapping of 5% silica-loaded Cu catalyst after CO₂RR at the region of interest two. Cu is in red and Si is in green.

Figure R9. HAADF-STEM image and EDX elemental mapping of 5% silica-loaded Cu catalyst after CO₂RR at the region of interest three. Cu is in red and Si is in green.

ii. In Figure 23 of the supplementary information it is indicated that the EDX was performed after the reaction in the area of Figure 2b, does it mean that an identical location study was performed? If this is the case, then the technical details of this experiment should be described.

The spectrum shown in Supplementary Fig. 23 is the integration of the spectra taken from the whole region shown in Fig. 2b. We presented it in the SI as validation for the chemical maps shown in Fig. 2c and d. The same electron beam that is used for STEM imaging rasters over the particles and the characteristic X-rays (resulting from electron-matter interaction) are collected by EDX detectors (four of them available in the microscope). This description is

added to the Methods section. For each EDX map, the area is exposed to the electron beam for 10-15 minutes, depending on the number of counts collected and the qualitative signal to noise ratio. Once the overall number of counts (detected X-rays) reached half a million (also depends on the area size and elements), the maps and the elemental signals were sufficiently clear.

We have included this description into our revised manuscript:

- **Page 16, Line 361:** “The microscope was equipped with four quadrant windowless silicon drift detectors EDX (Super-X) which increased the angle of collection, and hence, collection efficiency. During EDX mapping, the electron beam used for STEM imaging was rastered over the particles and the characteristic X-rays (resulting from electron-matter interaction) were collected by EDX detectors (four of them available in this microscope). For each EDX map, the area was exposed to the electron beam for 10-15 minutes, depending on the number of counts collected and the qualitative signal to noise ratio. In general, once the overall number of counts (detected X-rays) reached ~ 0.5M (depending somewhat on the area size and elements), the maps and the elemental signals were clear.”

iii. Supplementary Fig. 24 suggests that Si species are distributed on the Cu surface rather than in the Cu bulk., so some Si segregation occurs. In Figure 25 of the SI an homogeneous distribution of Si is reported, however it is not clear from which part of the particle in Fig 25a this is taken (bulk Si-free or Si-containing surface?. These data should be presented better. Also, in elemental mapping in Fig 25 c and d the pixel size seems much smaller than the actual resolution in Figure 25b. How small the step size can be in the mapping to integrate the signal? It looks to me more like a mapping of the noises rather than the signal. One would expect that the Cu signal would increase where the atoms are in Figure 25b. However atomically resolved EDX mapping is very difficult to achieve. But this shows also the technical flaw of this measurement. Are these measurements needed after all? I am getting back to the point of atomically dispersed clusters mentioned by the authors, what doe actually mean? Is it this all about excluding Si-Si direct bonding?

Fig. 2 indicates an overall homogeneous dispersion of silica on Cu. To identify the geometric location of silica clusters, a finer EDX mapping result in Supplementary Fig. 27 evidences the surface distribution of Si species rather than silica embedded in the bulk of Cu lattice.

Supplementary Fig. 25 was removed to prevent any potential confusion.

iv. Page 8 Line 183-191. The authors state that the increase in the silica loading to 5% diminishes the interaction between Si and Cu. Increasing and forms a strong Si-O-Si layer, destabilizing the Cu-SiO_x interface. This referee think that this should not be explained as diminishing of the interaction between Si-Cu, but a change from a 2D-thin layer of Si oxide into a 3-D overlayer.

We have revised the relevant statement in the revised manuscript:

- **Page 12, Line 258:** “From EXAFS fits, we also found a decrease in the Si-Cu coordination from 3 to 2 with the silica loading increasing from 2.5% to 5% (**Fig. 4e**), indicating that an increase in the silica loading to 5% diminishes the interaction between Cu and silica. Increasing the silica loading from 2.5% to 5% also results in the growth of silica molecules, evidenced by an increase in the Si-O coordination from 2 to 3, an increase in the Si-Si coordination from 1 to 2, and a ~7% contraction in the Si-Si bond from 3.0 Å to 2.8 Å (**Fig. 4e – f**). ”

Overall, this referee thinks that the paper is still not in a shape to be published in this journal. The nature of Si under in situ conditions and the effect of it on the Cu has to be clarified (electronic and geometric). If the Si role is to enhance the step sites this should be counted. Also the theoretical study is based on a mechanism of reaction proposed earlier and considering energetics of the intermediates. This referee thinks that the role of Si can be very different from what the authors are suggesting but other techniques more sensitive to light elements should be applied to uncover this.

The new in-situ Raman studies provide insight into the role of silica in facilitating CO₂ activation and reduction at the Cu-SiO_x interface. Both experimental and theoretical findings presented in the revised manuscript are in good agreement – strengthened by the insight offered by the Reviewer.

We thank the reviewer to help advance the mechanistic analysis of this work, and we hope that the revised work meets the standards for publication in Nature Communications.

Reviewer #2

I find the authors made much efforts in revision and supplemented more evidence for the raised questions. I believe the characterizations and the calculations represent the strong part. Additionally, I noticed in the experimental section, that the AEM was activated for at least 24 hours in 1 M KOH and rinsed with deionized (DI) water before test. Considering the etching feature of KOH to silica, I would ask whether rinsing is sufficient to remove the free KOH confined inside. Also I suggest the authors to discuss whether the enriching strategy of the interface sites by silica doping is a general way.

We have performed the rinsing process continuously for 15 minutes to remove the KOH from the surface and inside of the AEM.

We have now included further details of the rinsing process in the revised manuscript:

- **Page 18, Line 414:** “The AEM was activated for at least 24 hours in 1 M KOH and rinsed with deionized (DI) water continuously for 15 minutes to remove the KOH from the surface before the experiments.”

While we designed our experiments to remove KOH from the surface and inside the AEM, we have also performed a detailed literature review as well as DFT calculations to investigate this potential effect. According to our literature review, at room temperature, silica is stable at dilute KOH solutions (J. Electrochem. Soc. 137, 3612-3626 (1990)).

To show the universality of oxide modulation strategy, we theoretically constructed a Cu-GeO_x interface and assessed the role of GeO_x – an analogous oxide to silica – in affecting C-C coupling reaction with increasing GeO_x loading (**Figure R10-11**). We found a similar volcano-shaped dependence of the OCCOH* formation energy based on the GeO_x loading, in which the Cu-GeO_x catalyst with a GeO_x loading up to ~3% decreases the OCCOH* formation energy by ~-0.6 eV compared to the bare Cu catalyst.

We have included the new results and related discussions in the revised manuscript:

- **Page 14, Line 316:** “To demonstrate universality of our oxide modulation strategy, we theoretically assessed the role of GeO_x – an analogous oxide to silica – on the C-C coupling reaction at the Cu-GeO_x interface (**Supplementary Fig. 41 – 42**). By changing the GeO_x loading ranging from 0% to 5%, we find a similar volcano-shaped dependence of the OCCOH* formation energy based on the GeO_x loading. The Cu catalyst with a GeO_x loading up to ~3% decreases the OCCOH* formation energy by ~-0.6 eV compared to the bare Cu catalyst, which we attributed to the formation of the Ge-O bond between O^{δ-} of the OCCOH* intermediate and GeO_x at the Cu-GeO_x interface.”

Figure R10. DFT calculations of the Cu-GeO_x catalysts. a-c The energetically favorable Cu-GeO_x catalyst geometries at a GeO_x loading of 1.6% (a, 1/16 ML), 3.1% (b, 2/16 ML) and 4.7% (c, 3/16ML) over Cu(111). d The formation energy (ΔH_{rxn}) of OCCOH* ($CO^* + COH^* \rightarrow OCCOH^*$) over the pure Cu(111) and equilibrium Cu-GeO_x (blue sphere) and Cu-SiO_x (red sphere) catalysts; Volcano-shaped plots are achieved for the formation energies of OCCOH* over the equilibrium Cu-GeO_x and Cu-SiO_x catalyst geometries.

Figure R11. Calculations of the adsorption energies of CO+COH and OCCOH on different Cu catalysts. a-d The energy favourable CO_COH coadsorption over Cu(111) (a) and the Cu-GeO_x catalyst at a GeO_x loading of 1.6% (b), 3.1% (c) and 4.7% (d). **e-h** The energy favourable OCCOH adsorption over Cu(111) (e) and the Cu-GeO_x catalyst at a GeO_x loading of 1.6% (f), 3.1% (g) and 4.7% (h).

Reviewer #1 (Remarks to the Author):

The in situ RAMAN study is interesting and clarifies several aspects of this work. In contrast to the authors comment Si/N-C were investigated for CO₂ reduction (Liu, Y., Chen, S., Quan, X. & Yu, H. Efficient electrochemical reduction of carbon dioxide to acetate on nitrogen-doped nanodiamond. J. Am. Chem. Soc. 137, 11631–11636 (2015).).

This referee thinks that this work is now comprehensive and the authors have addressed every points satisfactorily.

Reviewer #2 (Remarks to the Author):

The authors made efforts, I do not oppose its publications. However, the following points still need to be altered/strengthened. In the aspect of catalyst design, i.e., creating effective Cu-SiO_x interfaces, it is not new, similar to ref 31, and ref 30 cannot support the author's claim that "silica has been used as an efficient adsorbent material to capture CO₂". Essentially, a narrow micropore and/or strong CO₂-philic sites as the KCl species in ref 30 contribute the most to the selective CO₂ trapping. In the aspect of electrolysis, the long-term stability at increased CO₂ concentration experienced relatively large fluctuations at around 5 hours. The reason should be clarified.

Manuscript ID: NCOMMS-20-33448B

“Silica-copper catalyst interfaces enable C-C coupling toward ethylene electrosynthesis”

Reviewer 1

The in situ RAMAN study is interesting and clarifies several aspects of this work. In contrast to the authors comment Si/N-C were investigated for CO₂ reduction (Liu, Y., Chen, S., Quan, X. & Yu, H. Efficient electrochemical reduction of carbon dioxide to acetate on nitrogen-doped nanodiamond. *J. Am. Chem. Soc.* 137, 11631–11636 (2015).).

This referee thinks that this work is now comprehensive and the authors have addressed every points satisfactorily.

We thank the reviewer for the comment. We carefully assessed the reference (*J. Am. Chem. Soc.* 137, 11631–11636 (2015)) indicated by the reviewer, in which Si nanorod was used as a substrate/template to coat N-doped nanodiamond catalyst for CO₂ reduction. Neither the role of Si, nor any influence of Si/N-C interactions on catalytic performance was discussed in this article.

Reviewer #2

The authors made efforts, I do not oppose its publications. However, the following points still need to be altered/strengthened.

In the aspect of catalyst design, i.e., creating effective Cu-SiO_x interfaces, it is not new, similar to ref 31, and ref 30 cannot support the author's claim that "silica has been used as an efficient adsorbent material to capture CO₂". Essentially, a narrow micropore and/or strong CO₂-philic sites as the KCl species in ref 30 contribute the most to the selective CO₂ trapping.

We have removed the reference to (*ACS Appl. Mater. Interfaces* **9**, 31683-31690 (2017)) in our revised manuscript.

In the aspect of electrolysis, the long-term stability at increased CO₂ concentration experienced relatively large fluctuations at around 5 hours. The reason should be clarified.

The system requires time to establish steady-state conditions with a balance of gas, ion, and water transports, particularly at high current densities. Thus fluctuations during the first 5 hours are attributed to the formation of stable cathode:AEM:anode interfaces at high CO₂ concentrations (40% and 100%).

We have clarified this point in our revised manuscript (**Page 14, Line 301**):

- “We noted fluctuations in the first 5 hours of stability testing with high CO₂ concentrations, a phenomenon we attribute to forming stable cathode:AEM:anode interfaces and stabilizing rates of gas, ion, and water transport within the MEA.”